Manuscript prepared for J. Name
with version 5.0 of the LATEX class copernicus.cls.
Date: 12 May 2016

# Simulated 2050 Aviation Radiative Forcing from Contrails and Aerosols

**Chih-Chieh Chen and Andrew Gettelman**

National Center for Atmospheric Research, P.O. Box 3000, Boulder CO 80307-3000, USA

*Correspondence to:* Chih-Chieh Chen, cchen@ucar.edu

**Abstract.** The radiative forcing from aviation induced cloudiness s investigated by using the Community Atmosphere Model Version 5 (CAM5) in the present (2006) and the future (through 2050). Global flight distance is projected to increase by a factor of 4 between 2006 and 2050. However, simulated contrail cirrus radiative forcing in 2050 can reach $87\,\mathrm{mW\,m^{-2}}$, an increase by a factor of 7 from 2006, and thus does not scale linearly with fuel emission mass. This is due to non-uniform regional increase in air traffic and different sensitivities for contrail radiative forcing in different regions.

Our simulations indicate that negative radiative forcing induced by the indirect effect of aviation sulfate aerosols on liquid clouds in 2050 can be as large as $-160\,\mathrm{mW\,m^{-2}}$, an increase by a factor of 4 from 2006. As a result, the net 2050 aviation radiative forcing has a cooling effect on the planet. Aviation sulfate aerosols emitted at cruise altitude can be transported down to the lower troposphere, increasing the aerosol concentration, thus increasing the cloud drop number concentration and persistence of low-level clouds. Aviation black carbon aerosols produce a negligible net forcing globally in 2006 and 2050.

Uncertainties in the methodology and the modeling are significant and discussed in detail. Nevertheless, The projected percentage increase in contrail radiative forcing is important for future aviation impacts. In addition, the role of aviation aerosols in the cloud nucleation processes can greatly influence on the simulated radiative forcing from aircraft induced cloudiness, and even change its sign. Future research to confirm these results is necessary.

**Keywords.** contrail, climate modeling, radiative forcing, aviation aerosols, aviation impact

## 1 Introduction

Aviation fossil fuel consumption is projected to increase significantly during the 21st century due to population and economic growth. Thus the aviation impact on climate is expected to intensify substantially from its current estimated level (IPCC, 1999; Lee et al., 2009) faster than other sources, and become a larger component of future radiative forcing than it is today. Since future population and economic growth are most likely non-uniform, some regions may experience higher aviation impact than others. For example, with the recent rapid development in Asia during the past decades, this region is likely to join central Europe and the eastern US with high regional aviation emissions.

Aviation causes several important climate impacts. Linear contrails form when aircraft exhaust mixes with ambient air that is cold and moist enough (Schmidt, 1941; Appleman, 1953). Subsequently, spreading and shearing of contrails may increase cloudiness, called contrail cirrus (Schumann and Wendling, 1990; Minnis et al., 1998). Linear contrails and contrail cirrus produce a warming effect on the planet since the radiative forcing of these optically thin high clouds is dominated by longwave heating (Dietmüller et al., 2008; Rap et al., 2010; Kärcher et al., 2010; De Leon et al., 2012). Nevertheless, it is a challenge to quantify the radiative forcing of contrail cirrus since this requires accurately accounting for the spreading of contrails. Based on some recent studies using general circulation model simulations, Burkhardt and Kärcher (2011) estimated present day contrail cirrus radiative forcing of $31\,\mathrm{mW\,m^{-2}}$, Chen and Gettelman (2013) estimated $13\,\mathrm{mW\,m^{-2}}$, and Schumann and Graf (2013) estimated $50\,\mathrm{mW\,m^{-2}}$. More recently, Schumann et al. (2015) estimated $60\ \mathrm{mW\,m^{-2}}$.

Aircraft also emit various aerosols, such as sulfate and Black Carbon (BC, or soot). In addition to directly absorbing and reflecting radiation, sulfate and BC can alter cloud prop-

erties, such as drop and crystal concentration, and cloudiness in the troposphere. These "indirect effects" of aviation aerosols may result in a change in cloud radiative forcing. Hendricks et al. (2005, 2011) found that aviation BC could significantly increase the crystal concentration if aviation BC could serve efficiently as ice nuclei. Furthermore, Penner et al. (2009) reported, by assuming aviation BC as highly efficient heterogeneous ice nuclei, that aviation BC could induce an indirect forcing ranging between $-161$ and $+25$ $\mathrm{mW\,m^{-2}}$ under different sensitivity tests. Liu et al. (2009) also found a similar range for the forcing of aviation BC with high uncertainty depending on the assumptions made. However, using a more typical ice nucleating efficiency of BC (DeMott et al., 2009, 2010, 0.1 %) and size distribution for aviation BC, Gettelman and Chen (2013) found negligible direct and indirect BC radiative forcing. Aviation sulfate aerosols, however, have been estimated to be producing at present a radiative forcing of $-46\,\mathrm{mW\,m^{-2}}$, by altering the properties of warm clouds (Gettelman and Chen, 2013), e.g. drop concentration and liquid water path. Similar effects of aviation sulfate aerosols were reported by Righi et al. (2013). These values are larger than the potential warming effect of contrails and contrail cirrus.

In this study, we will explore the future aviation impact on climate by employing a comprehensive general circulation model and four different future scenarios. We will focus on the radiative forcing induced by contrail cirrus and aviation aerosols: by far the most uncertain aviation radiative forcing components (Lee et al., 2009). Model description and future aviation emission scenarios are described in Sect. 2, model simulation results are presented in Sect. 3, and discussions are in Sect. 4.

## 2  Methodology

### 2.1  Model description

Community Atmosphere Model version 5 (CAM5) is employed in this study and a detailed scientific description can be found in Neale et al. (2010). The model includes a detailed treatment of liquid and ice cloud microphysics (Morrison and Gettelman, 2008), including a representation of particle size distributions, a detailed representation of mixed phase clouds with the consideration of water uptake onto ice (the Bergeron–Findeisen process) and ice supersaturation (Gettelman et al., 2010). This is coupled to a consistent radiative treatment of ice clouds, and an aerosol model (7-mode aerosol model employed in this study) that includes particle effects on liquid and ice clouds (Liu et al., 2012). This method has the advantage of being self-consistent with the climate simulation.

The evolution of BC and sulfate aerosols in CAM5 is described in detail in Liu et al. (2012). Briefly, BC is emitted to the primary carbon mode, then is aged into the accu-

mulation mode by condensation of $H_2SO_4$, $NH_3$ and semi-volatile organics and by coagulation with Aitken and accumulation modes. Sulfate aerosols are emitted to Aitken mode, and is aged into the accumulation mode through coagulation and condensation. Within each mode aerosols are internally mixed and the optical properties reflect this. Aviation aerosols are treated in the same manner, and simply added to the existing modal description. Activation of aerosols to ice nuclei is calculated by the activation scheme following Liu et al. (2009); Gettelman et al. (2010), with homogeneous freezing on sulfate. It is assumed that 0.1 % of black carbon can be activated as heterogeneous ice nuclei (Gettelman and Chen, 2013).

As shown in Chen et al. (2012), CAM5 is capable of simulating the mean relative humidity and reproducing the distribution of the frequency of ice supersaturation in the upper troposphere and lower stratosphere (UTLS) as observed from the Atmospheric Infra Red Sounder (AIRS) satellite (Gettelman et al., 2006). These attributes are critical in simulating contrails.

The contrail parameterization used in this study is described in detail in Chen et al. (2012). The parameterization follows the Schmidt–Appleman Criteria (Schmidt, 1941; Appleman, 1953) to determine if aviation water emissions should be vapor or condensed water (ice) depending on the ambient conditions. The implementation follows Ponater et al. (2002): contrails are initialized if aviation water vapor encounters an ambient condition with the atmospheric temperature below a critical temperature, as a function of atmospheric pressure, and the ambient humidity above ice supersaturation. Otherwise, the aviation water emission is added to the background water vapor.

We use the approximation of the critical temperature ($T_c$ in °C) for contrail formation given by Schumann (1996)

$$T_c = -46.46 + 9.43\ln(G - 0.053) + 0.72[\ln(G - 0.053)]^2,$$

and $G$ in the units of $\mathrm{Pa\,K^{-1}}$ is defined as

$$G = \frac{\mathrm{EI_{H_2O}} \cdot c_p \cdot p}{\varepsilon Q(1 - \eta)},$$

where $\mathrm{EI_{H_2O}}$ is the emission index of water vapor, $c_p$ the specific heat of air at constant pressure, $p$ the atmospheric pressure, $\varepsilon$ ratio of the molecular weight of water and air, $Q$ the specific combustion heat, $\eta$ the propulsion efficiency of the jet engine. In this study, $\mathrm{EI_{H_2O}} = 1.21$ ($\mathrm{kg\,H_2O\,kg^{-1}\,fuel}$), $Q = 4.3 \times 10^7\,\mathrm{J\,kg^{-1}}$, $\eta = 0.3$.

One important assumption made in this parameterization is that the initial volume of contrails is a product of the flight path distance and a cross-sectional area, assumed to be $300\,\mathrm{m} \times 300\,\mathrm{m}$ (Chen et al., 2012). The ambient humidity above ice saturation within this volume is assumed to become part of the contrail ice mass. Ice particles within contrails when initialized are assumed to be spherical and have an initial diameter of $10\,\mathrm{\mu m}$ (Schröder et al., 2000). Chen

et al. (2012) discuss the sensitivity of contrail forcing to the choice of particle size and cross-sectional area. When a contrail is initialized, its cloud fraction is calculated by its volume and that of the grid box and we assume there is no overlap with the existing clouds within the grid box. After contrails are initialized, they become indistinguishable from the background cloud field, i.e. there is no separate cloud type for contrails in CAM5, and they evolve in the same manner as all clouds in CAM5.

We emphasize that this parameterization is designed to consistently estimate the climate impact of aviation emissions in a manner consistent with the cloud representation in a global climate model. The method parameterizes the initial properties of contrails in a coarse and semi empirical way (Chen et al., 2012). Its utility is a consistent mass-conserving treatment of the effects of contrails in a global context. Uncertainties are significant, but have been treated parametrically in previous work noted above. We have noted significant uncertainties in the model in relation to the effects here as appropriate below.

## 2.2 Future aviation emissions scenarios

Future aviation emission scenarios have been discussed in IPCC (1999); Gierens et al. (1999); Marquart et al. (2003). In this study, four future aviation emission scenarios of identical flight tracks and flight distance are considered in this study, along with a present day scenario. Scenarios were developed by (Barrett et al., 2010) and are listed in Table 1.

Based on projected population and economic growth by 2050 with the current aviation technology, a "baseline" aviation emission scenario is obtained (BL). Under this scenario, the annual global fuel burn is $1.1 \times 10^{12}$ kg and $SO_4$ emission is $4 \times 10^7$ kg in 2050. Scenario 1 (SC1) assumes the same 2050 flight distance with an assumption of 2 % gain in engine efficiency per year, which reduces fuel consumption with the same flight distance as BL. Scenario 2 (SC2) is obtained with SC1 fuel consumption by assuming an alternative fuel with no sulfate emissions and 50 % reduction in aviation BC emissions. Scenario 3 (SC3) is the same as Scenario 2 except an increase of 5 % in water vapor emission is assumed. Future aviation emissions based on these four scenarios are available for the years of 2016, 2026, 2036, and 2050. It is important to emphasize that the flight distance for any given year remains the same under all four scenarios, and just the fuel use varies.

As illustrated in Fig. 1, the global flight distance is projected to increase by a factor of 4 from 2006 to 2050 and the fuel consumption is projected to increase by a factor of 5 under BL and 2.7 under SC1 by 2050. The biggest increase in flight distance (a factor of 6 by 2050) is found in East Asia, due to the projected rapid economic growth in this region. The fractional increase in flight distance in central Europe is nearly the same as that of the global flight distance, while

the increase in eastern US (a factor of 2.5 in 2050) is much lower.

## 2.3 Modeling framework and background meteorology

The GCM results are sensitive to vertical resolution (Chen et al., 2012), so we use a vertical resolution consistent with the design of CAM5. Following the methodology described in Gettelman and Chen (2013); Chen and Gettelman (2013), CAM5 with specified dynamics (CAM5-SD) is employed for this study. Future background meteorology is obtained from fully coupled simulations using Community Earth System Model (CESM) under Representative Concentration Pathways (RCPs) with radiative forcing in 2100 of 4.5 and 8.5 $W m^{-2}$ (van Vuuren et al., 2011), hereafter RCP4.5 and RCP8.5. Future emissions from non-aviation sources and greenhouse gas concentration are based on RCPs for the respective year of the aviation emissions for CAM5-SD simulations. RCP8.5 is selected for this study because the current global progression fits the trajectory of RCP8.5 most closely. To examine the sensitivity of the results under a different future scenario, a lower emission scenario RCP4.5 is selected. To address the uncertainty due to background meteorology, we use four different annual cycles from a CESM coupled experiment with a given RCP to drive CAM5-SD and perform 20 year SD simulations (repeating an annual cycle each year). For example, for 2050 aviation emissions, we use the background meteorologies from years 2049, 2050, 2051, and 2052 to drive CAM5-SD to form an ensemble. This modeling strategy is employed instead of transient simulations because Chen and Gettelman (2013) found that the model variability of CAM5 was above the expected radiative forcing of aviation emissions when it was in a free running mode which would not allow us to effectively obtain forcing of statistical significance. The radiative forcing is then estimated from 80 simulated years. All simulations presented in this study are run on a $1.9° \times 2.5°$ latitude–longitude grid with 30 vertical layers.

Due to global warming, atmospheric temperature at the cruise altitude is warmer under both RCP4.5 and RCP8.5 than present day meteorology. Note also that there are slight shifts in flight altitude as newer planes fly at higher altitude which is about 20 hPa lower in atmospheric pressure in 2050 than in 2006. The upper tropospheric warming reduces the frequency for contrail formation (Fig. 2). A potential contrail can form when the temperature and humidity satisfy the Schmidt–Appleman contrail formation criteria. Furthermore, if the atmosphere is supersaturated with respect to ice, the contrail will persist. The critical threshold temperature is the dominant factor in the Schmidt–Appleman Criteria (Schmidt, 1941; Appleman, 1953). Figure 2b and c indicate that future atmospheric conditions in the tropics and subtropics is less favorable for contrail formation at the cruise altitude, mainly due to higher atmospheric temperatures. The reduction in the frequency of persistent contrails over East

Asia (Fig. 2e and f) has important implications since there is substantial projected increase in air traffic in this region. The reductions are larger in RCP8.5 than in RCP4.5 due to stronger warming. In mid and high latitudes at cruise altitude, however, future atmospheric conditions become more favorable for contrail formation (Fig. 2b and c). Figure 2e and f indicate that the frequency of persistent contrails in 2050 is increased slightly over eastern US and central Europe.

## 3   Results

Three sets of simulations are conducted in this study. The first set is the control simulation in which no aviation emissions are included. The second set of simulations include aviation water vapor emission only and thus the difference from the control run can be interpreted as the effect of contrails and contrail cirrus ("Contrail Cirrus" or "$H_2O$"). The third set of simulations include aviation water vapor, sulfate, and BC emissions, and thus the difference from the control run is due to the combination of contrail cirrus and aviation aerosols ("$BC + SO_4 + H_2O$"). Hence, the impact of aviation aerosols may be deduced by taking the difference between the second and third sets of simulations.

### 3.1   *Global Average*

The radiative forcing of contrail cirrus in 2050 simulated by CAM5 ranges between 75 and 95 $mW\,m^{-2}$ based on various aviation emission scenarios and different background meteorology (Fig. 3a). Based on our simulations, the highest radiative forcing of contrail cirrus (red dashed line in Fig. 3a) is reached when BL emission (higher fuel consumption) is incorporated with present-day meteorology. In other words, the effect of global warming, i.e. background meteorology under both RCPs, is to reduce radiative forcing of contrail cirrus. Conversely, the lowest radiative forcing (green solid line in Fig. 3a) can be obtained when SC1 emission (lower fuel consumption) is incorporated with background meteorology under RCP8.5 (warmest ambient temperatures at cruise altitude).

Figure 3a and Table 3 indicate that when considering global contrail cirrus RF only (no aerosol effects), the spread across emissions scenarios in 2050 is 73–83 $mW\,m^{-2}$ (13 %, based on 2050 RCP8.5 meteorology), while the spread due to future meteorology (RCP4.5 v. RCP8.5) in 2050 is only 80–87 $mW\,m^{-2}$ (8 %, based on 2050BL), thus the spread in scenarios (and total fuel burn) is more important than the climate scenarios (the degree of projected warming). Warmer worlds (RCP8.5) result in less contrail cirrus RF.

Our results (Table 3) indicate that globally-averaged contrail RF in 2050 is projected to increase by a factor of 6 to 7 from 2006 which is not strongly affected by the background meteorology (RCP4.5 vs RCP8.5) and emissions scenarios (BL, SC1, SC2, SC3). This is surprising since the global avi-

ation fuel consumption is projected to only increase by a factor of 4.8 under BL and 2.7 under SC1 from the 2006 level. Notice that the increase in contrail RF is not uniform globally. As indicated in Table 3, the percentage increase in contrail RF from 2006 to 2050 is most pronounced in East Asia which is consistent with the projected fuel consumption (see Fig. 1). Since East Asia is lower in latitude than East US and Central Europe, the increase in contrail RF will carry even higher weighting when computing the global average. Thus, the contrail RF in East Asia is an important contributor of the globally-averaged contrail RF in 2050.

Another unexpected result is the minimal spread in contrail RF among different emission scenarios (Table 3). In 2050, the fuel consumption is projected to increase by a factor of 4.8 from the 2006 level under BL and only 2.7 under SC1. However, these two scenarios produce very similar contrail RF. Even though fuel consumption is very different under these two scenarios, the flight distance is identical (see Fig. 1). Under our contrail parameterization, the initial volume of contrails is dependent on the flight distance which determines the uptake of the ambient water vapor for the formation of contrails. The minimal spread in contrail RF among different emission scenarios implies that the uptake of the ambient water vapor is a major portion of the contrail ice mass.

Figure 3b illustrates the combined radiative forcing of contrail cirrus and aviation aerosols. Clearly, the net effect is positive when aviation sulfate emission is eliminated (SC2 and SC3, blue and magenta lines in Fig. 3b), and these scenarios are nearly identical to the SC1 and SC3 contrail only results in Fig. 3a. The corresponding deduced aerosol forcing (blue and magenta lines in Fig. 3c), i.e. aviation BC, is nearly zero. Hence, the negative forcing is produced by aviation sulfate aerosols. Figure 3b illustrates that the sum of the negative forcing induced by aviation sulfate aerosols, aviation BC and contrail cirrus heating is negative, consistent with the findings by Gettelman and Chen (2013). The effects are larger (more cooling) for the 2050 BL case (red) than for SC1 (green), mostly due to the larger RF from aerosols (Fig. 3c). 5 % additional aviation water vapor emission is not significant in enhancing contrail radiative forcing (see Fig. 3), since the magenta SC3 lines with enhanced aviation water are not significantly different from SC2. Note that uncertainties for these findings can result from the assumptions on the particle size of aviation aerosols and the fraction of efficient ice nuclei for aviation BC, as shown in Gettelman and Chen (2013).

### 3.2   *Zonal Mean Perturbations*

Figure 4 presents zonally averaged perturbations due to 2050 aviation emissions with 2050 meteorology under RCP8.5, highlighting the difference between the effects of contrails through aviation emissions of water ("$H_2O$": green in Fig. 4) and the effects of water and aerosols ($H_2O + SO_4 + BC$: blue in Fig. 4). The effect of water is only significant when

the water condenses as contrail or contrail cirrus: the direct RF of aviation water vapor is not significant, as seen in Fig. 3 and noted by Chen et al. (2012). Aviation aerosols are found to enhance both the negative shortwave cloud forcing (Fig. 4a) and positive longwave cloud forcing (Fig. 4b), with a larger effect on the shortwave. The net forcing of aviation becomes mostly negative (Fig. 4c). The aerosol effects on the cloud forcing arise mainly through enhancing cloud drop number concentration (Fig. 4e) and liquid water path (Fig. 4g), which are not significantly affected by aviation $H_2O$ forming contrails. Cloud top ice number concentration increases by aviation are enhanced with aerosols (Fig. 4f) which in turn raises ice water path more than water emissions do alone. As expected, these effects are mainly found in the mid and high latitudes in the Northern Hemisphere where the most significant aviation emissions occur. As seen in the global averages (Fig. 3c), for SC2, with no sulfur, aviation BC produces a negligible radiative forcing, and thus most of the negative radiative forcing is due to aviation sulfate aerosols.

Perturbations produced by different aviation emissions scenarios are shown in Fig. 5. These simulations include aviation water vapor, sulfate and BC emissions. SC3 is not significantly different from SC2, so it is not shown. Baseline emissions (blue) are the same in Fig. 4 and Fig. 5. The largest magnitude shortwave and longwave cloud forcings, are produced by the 2050 BL (largest fuel consumption). However, the largest net cloud forcing (Fig. 5c), is produced by the 2050 Scenario 2 emissions, which has a lower amount of fuel consumption than the 2050 BL. But SC2 has no sulfate emissions, and so the cooling effect resulting from aviation sulfate aerosols (in the shortwave cloud radiative effect, Fig. 5a) is removed.

The effect of aviation sulfate aerosols is readily detected by examining the characteristics of warm clouds. 2050 SC2 emissions produce minimal change in the liquid cloud drop number concentration (Fig. 5e) and liquid water path (Fig. 5g) which are both close to the 2006 aviation emissions level. The perturbations in the ice cloud number (Fig. 5f) and ice water path (Fig. 5h), on the other hand, are clearly related to the fuel consumption in each emission scenario.

It is interesting to note that the total cloud cover is reduced in the mid latitudes of the Northern Hemisphere as shown in Figs. 4d and 5d where contrails are most frequently formed. The formation of contrails should increase the cloud fraction and its volume is, under our contrail parameterization, the product of flight distance and a cross-section of 300 m and 300 m. However, due to the uptake of background humidity upon the formation of contrails which leads to a reduction of relative humidity of the grid cell. Such dehydration process has been discussed in Brasseur et al. (1998, 2015); Sausen et al. (2005); Lee et al. (2009, 2010); Boucher et al. (2013). Indeed, it is found that the relative/specific humidity at the cruise altitude is slightly reduced over Central Europe and Eastern US (not shown). Since the relative humidity of a grid cell is an important factor for the model cloud scheme to determine the cloud fraction, this effect is to reduce the cloud fraction. Therefore, these two factors constantly compete to determine the net change in cloud fraction. As illustrated in Fig. 5d, based on the 2006 emission level the first factor is more important and thus it produces an increased in cloud cover. In 2050, however, the second factor overwhelms the first.

### 3.3 *Regional radiative forcing*

Figure 6 illustrates the top of the atmosphere radiative forcing (as the change in the net residual flux at the top of model: RESTOM) in $W m^{-2}$ for 2050. Figure 6a,b,c includes only cirrus (contrail) effects from $H_2O$ and Fig. 6d,e,f includes cirrus and aerosol effects. Similar as in 2006 (Gettelman and Chen, 2013), 2050 radiative forcing is largest over central Europe and eastern North America. Including aviation aerosols makes the aviation forcing perturbation negative over the oceans Cooling is more efficient over the ocean due to low albedo from a dark ocean surface.

We focus quantitatively on several key regions, including eastern North America, central Europe, and East Asia. The exact boundaries in these three regions can be found in Table 2. The regions are defined and detailed in Fig. 6, and quantitative estimates of the forcing in each region for the different scenarios are in Table 3.

Contrail cirrus radiative forcing over central Europe in 2050 is projected locally reach 2 $W m^{-2}$ (Fig. 6c), which is equivalent to a factor of 2–3 increase from the 2006 level (Table 3). The contrail cirrus radiative forcing over the eastern US (Fig. 6a) could reach $800\,mW m^{-2}$ in 2050 and the 2050 increase over 2006 forcing is even higher percentage-wise (Table 3). Nevertheless, the most dramatic increase in contrail cirrus radiative forcing is found in East Asia (an increase of 600 % in 2050BL). This feature is consistent with this region having the largest projected increase in aviation fuel consumption.

When aviation aerosols are also included in the simulations, our simulations indicate that the positive forcing over land is slightly reduced from the forcing induced by contrail cirrus alone (Fig. 6d–f). Similarly as found in Gettelman and Chen (2013) based on the 2006 emissions, the negative forcing induced by aviation emissions is mainly found over the ocean due to the the low surface albedo. The perturbation is larger over the oceans also because the environment is cleaner (less aerosols) than over land. Over the three regions with the highest air traffic projected in 2050, i.e. eastern US, central Europe, and East Asia, aviation aerosols reduce the regionally-averaged positive radiative forcing induced by contrail cirrus by roughly 50 %, as shown within the blue boxes in Fig. 6. The peak positive forcing within each of the three regions is also reduced by roughly 50 % due to aviation aerosols. Regional estimates are provided in Table 3.

The negative radiative forcing by aviation aerosols found in this study is similar as in Righi et al. (2013). The intensity of the cooling is likely to be dependent on the background cloud drop number concentration.

### 3.4  *Seasonal aviation impacts*

Thus far the focus has been annual means, but there is also a seasonal cycle to the aviation radiative forcing. The 2050 contrail cirrus radiative forcing under three aviation emission scenarios exhibits a similar annual cycle (Fig. 7g) as that with the 2006 aviation emission level (Chen and Gettelman , 2013), i.e. higher forcing during the winter months and lower forcing during the summer months in the Northern Hemisphere. The seasonality results from the colder atmospheric temperature in the winter months favoring the formation of contrails (Chen and Gettelman , 2013). The amplitude of the annual cycle of forcing in 2050 is larger than 2006, due to higher aviation emissions in 2050. The three aviation emission scenarios produce similar contrail cirrus radiative forcing, but consistent with Fig. 3a, the baseline scenario has about 10–15 % larger magnitude over the annual cycle. When examining shortwave and longwave radiative forcing separately, however, shortwave radiative forcing reaches its minimum in April (Fig. 7a) and longwave radiative forcing has two maxima in April and November (Fig. 7d).

When aviation aerosols are incorporated in our simulations with aviation water vapor emissions, the magnitude of both the shortwave and longwave radiative forcing become larger (compare Fig. 7b and e with Fig. 7a and d). Note that the short wave is a larger negative number. The net radiative forcing (Fig. 7h) becomes negative except during the winter months in the Northern Hemisphere in the BL and SC1. In SC2 and SC3 emissions in which there is no sulfate emission, the annual cycle of radiative forcing resembles that of contrail cirrus in Fig. 7g. The stronger induced shortwave forcing in the summer is mainly due to more intense incident solar radiation.

The effect of aviation aerosols is deduced by taking the difference between the two sets of simulations, i.e. the first set which includes only aviation water vapor emissions and the second set which includes both aviation water vapor and aerosol emissions. Since aviation sulfate aerosols are eliminated in SC2 and SC3, the net effect is due to aviation BC alone (half of SC1). It is found that aviation BC produces a slight negative shortwave forcing (Fig. 7c) and positive longwave forcing (Fig. 7f), and its net forcing is nearly zero (Fig. 7i) with a global average of $-4\,\mathrm{mW\,m^{-2}}$ for SC2 and 0 for SC3 (see Tables 3 and 4). Thus, the much stronger forcing found in BL and SC1 is due to the presence of aviation sulfate aerosols. While the longwave forcing exhibits little annual variation (Fig. 7f), the shortwave forcing has a strong annual cycle (Fig. 7c), with the strongest negative forcing during the summer months in the Northern Hemisphere. As

illustrated in Fig. 7i, the net forcing of aviation aerosols is mainly controlled by the shortwave.

The signature of contrail cirrus may be readily detected by taking the difference in ice water path (IWP) between the simulation with aviation water vapor emissions and that without (Fig. 8a and b). The increase in IWP is mostly found in regions with high air traffic density, up to $4\,\mathrm{g\,m^{-2}}$. With the background IWP over central Europe and eastern US in the range of $40\,\mathrm{g\,m^{-2}}$ in June–July–August (JJA) and $20\,\mathrm{g\,m^{-2}}$ in December–January–February (DJF), the increase of IWP due to contrail cirrus in 2050 is about 10 % in JJA and 20 % in DJF over these regions. The results also indicate that the enhancement in IWP due to the presence of contrail cirrus is higher during the winter months in the Northern Hemisphere (Fig. 8b) than the summer months (Fig. 8a). This is due to the effect of colder atmospheric temperature in the upper troposphere during the winter which favors the formation of contrails (Chen and Gettelman , 2013). The inclusion of aviation aerosols is found to further increase IWP (Fig. 8c and d). The deduced effect on IWP by aviation aerosols is illustrated in Fig. 8e and f by taking the difference between simulations with aerosols and contrails, and just contrails. Such enhancement in IWP is mainly due to aviation BC since homogeneous freezing due to sulfate aerosols is not effective. The similarity of the enhancement in IWP between SC1 (green line) and SC2 (red line) illustrated in Fig. 5h reveals that aviation sulfate aerosols have a minimal effect on IWP (recall SC2 eliminates all sulfate emissions). In addition to enhancing IWP where contrail cirrus is present (Fig. 8a and b), the effect of aviation aerosols can spread over high latitudes and into polar regions. This is clearly due to atmospheric transport processes and the lower atmospheric temperature near the cruise altitudes in high latitudes that favors the formation of ice clouds. Note that contrail cirrus forms in flight corridors and a substantial portion of its ice mass originates from the water vapor of the model hydrologic cycle. But aviation aerosols have impacts over their life cycle in the simulations, i.e. they may produce perturbations far from flight corridors since they can be advected long distances.

Sulfate aerosols are highly efficient cloud liquid condensation nuclei, and can enhance liquid cloud drop number and increase liquid water path (LWP) by reducing sedimentation and precipitation as water remains in a larger number of smaller particles. Figure 9 illustrates the mean LWP (Fig. 9a and b) and the effects of aviation water and aerosols. Since contrails have minimal impact on LWP (Fig. 4g), the increase in LWP is mainly due to aviation aerosols. Furthermore, SC2 which eliminates sulfate emissions produces minimal increase in LWP (red line in Fig. 4g), implying most of the increase in LWP is due to aviation sulfate aerosols (compare SC1 and SC2 in Fig. 4g). Note that LWP is found at altitudes well *below* most significant aviation emissions, especially over the oceans. Water only ("contrail cirrus") simulations indicate no significant change in LWP (e.g. Fig. 3g)

indicating that aviation water and contrails do not alter LWP. LWP is only enhanced by aviation sulfate aerosol emissions in the simulations (Fig. 9c and d). It is found that largest increase in LWP is typically found where the background LWP and air traffic is large (compare Fig. 9c and d with Fig. 9a and b), which implies that the aviation sulfate aerosols mainly modify the properties of existing clouds instead of creating new clouds. Aviation aerosols emitted at cruise altitude can be transported down to near Earth's surface and thus the aerosol concentration in the lower troposphere can be substantially increased in remote regions. The increase of Aitken sulfate mass burden in flight regions over the ocean can reach 20 % and over Eastern US and Central Europe it is nearly 50%, in 2050 based on BL emissions. This increase in aerosols in turn raises the cloud drop number concentration of low-level clouds, by about 10 % based on 2050 emission levels. This brightens the existing low-level clouds, known as the Towmey effect (Twomey, 1977). Furthermore, the low-level clouds are found to be more persistent with around 20 % higher in lifetime as indicated by the model variable which records the frequency of the presence of clouds (not shown), due to the presence of aviation sulfate aerosols, known as the Albrecht effect (Albrecht, 1989). This chain of effects has several uncertainties that need to further evaluated. The first is whether aerosol burdens in flight regions increase is due to aircraft. The second is whether the model indirect effects are of the right magnitude. Both effects appear large, and thus this work could be treated as a high estimate.

We investigated whether increased aerosol concentration near the Earth's surface are due to aviation emissions transported from the cruise altitude. An experiment was conducted in which all aviation emission below 500 hPa was eliminated in the simulation and the result was nearly identical. This implied that vertical transport of aviation emissions from the cruise altitude down to the lower troposphere was responsible for the observed low-level cloud brightening, but not the aviation emissions during takeoff and landing. Similar results were reported in Barrett et al. (2010b). In addition to atmospheric transport processes, the distribution of sulfate aerosols can be affected by wet scavenging in the model as suggested in Liu et al. (2012). Thus, future work will be conducted in varying wet scavenging in the model to address its uncertainty.

The modification of cloud properties due to the presence of aviation aerosols also generates significant radiative forcing (Fig. 10). The striking similarity in pattern between Figs. 10a and 9c implies that the shortwave radiative forcing due to aviation aerosols is mostly induced by the low-level increases in LWP. The agreement between Figs. 10b and 9d is mainly observed in the tropics because mid and high latitudes do not receive much solar radiation during the winter months in the Northern Hemisphere. The negative shortwave radiative forcing is thus mainly produced by aviation sulfate aerosols since it is in excellent agreement with the increase of LWP.

### 3.5 *Aviation BC*

Aviation BC can also modify cloud properties. By comparing the red line (SC2) in Fig. 5g (no Sulfur, 50 % BC) and the green line (BL) in Fig. 4g (no Sulfur or BC), one may conclude that the presence of aviation BC may enhance LWP slightly. Similarly a comparison between Figs. 5h and 4h reveals that aviation BC also enhances IWP in the northern polar regions. Notice the difference in IWP in the Northern polar region between the green curve in Fig. 4h (2050 BL, $H_2O$ only) and the red curve in Fig. 5h (2050 SC2, $BC + H_2O$).

To examine the radiative forcing of aviation BC, an experiment with only BC emissions based on 2050 BL was performed and the radiative forcing is illustrated in Fig. 11. The agreement between the two panels indicate that the radiative fluxes at the top of the atmosphere induced by aviation BC is mainly attributed to cloud forcing (indirect effects). The net forcing of aviation BC is positive in high latitudes and polar regions in the Northern Hemisphere, and is negative in the tropics. These effects cancel out and yield nearly zero forcing globally.

Nevertheless, it is widely accepted that the impact of aviation BC is highly uncertain based on the assumptions made on its ice nucleation efficiency. By assuming high heterogeneous ice nucleation efficiency for aviation BC, Penner et al. (2009); Liu et al. (2009) found that aviation BC could produce radiative forcing of a wide range, either positive or negative, under different sensitivity tests. More recently, Zhou and Penner (2014) simulated large aviation BC induced radiative forcing when preactivation of aviation soot is assumed. Therefore, the radiative forcing of aviation BC is highly uncertain and this remains an active area of research.

### 4 Summary and conclusions

Four aviation emission scenarios were considered in this study to examine the aviation impact on climate through 2050. The global flight distance is projected to increase by a factor of 4 by 2050 from the 2006 level in all scenarios. The global radiative forcing for contrail cirrus in 2050 level can reach $87\,\mathrm{mW\,m^{-2}}$ which represents a factorial increase of 7 from 2006. Global warming due to anthropogenic greenhouse gases during the 21st century was found to reduce the intensity of contrail cirrus radiative forcing through suppressing the formation of contrails as the upper troposphere gets warmer. Note that the upper tropospheric relative humidity remains nearly constant as the planet warms. Thus stronger warming (in the RCP8.5 scenario) in 2050 results in a larger reduction in contrails than in the RCP4.5 scenario. This result is qualitatively robust across all the aviation scenarios examined.

The aviation emission scenario 1 (SC1), assuming a 2 % gain in engine efficiency per year (i.e. less fuel consumption), was found to have a very minor influence on the con-

trail cirrus radiative forcing compared to the 2050 BL case. Figure 1 indicates that there is roughly 40 % reduction in fuel consumption in 2050 under SC1 compared with BL, but the reduction of contrail radiative forcing is only about 13 % (see Table 3). This finding indicates that contrail cirrus radiative forcing does not scale linearly with emission mass which is mainly due to different sensitivities in different regions. Our results also suggest that it is unlikely that the intensity of positive contrail cirrus radiative forcing could be reduced with improvements of engine efficiency.

SC1 with reduced fuel usage, however, significantly reduced the negative forcing induced by aviation sulfate aerosols (Fig. 3b). Nevertheless, even with significant improvements of engine efficiency under SC1, the aerosol cooling still dominates warming by contrail cirrus.

The simulations indicate that SC3, with 5 % increase in aviation water vapor emissions from SC1, produced no significantly different contrail radiative forcing. This is due to the fact that a major portion of ice mass in contrails is from the uptake of water in the ambient atmosphere. However, a greater spread for the radiative forcing for contrail cirrus among BL, SC1, SC2 can be obtained if the propulsion efficiency $\eta$, held as 0.3 for all simulations in this study, is adjusted to reflect a change in fuel efficiency (Schumann, 1996) for different emission scenarios.

We need to emphasize, however, that the present-day radiative forcing of contrail cirrus simulated by CAM5 Chen and Gettelman (2013) is lower compared with several other studies Burkhardt and Kärcher (2011); Schumann and Graf (2013). This is likely due to the treatment of contrails in our contrail parameterization, i.e. there is no separate treatment of contrails from the background clouds upon formation. The ice particle size of young and mature contrails could be significantly different from natural cirrus and this could produce very different radiative forcing. In this study, we assume an initial ice particle size of $10\,\mu\text{m}$ due to the 30-minute time step of our simulations. For fresh and young contrails, the particle size can be significantly smaller which will increase the ice number concentration and optical thickness of the contrails substantially. Furthermore, our parameterization does not consider direct participation of aviation BC during the formation process of contrails which could highly affect the ice particle size and number concentration of fresh contrails. Nevertheless, the projected percentage increase of contrail radiative forcing shown in this study could be used to extrapolate the future contrail radiative forcing from other studies.

Our simulations indicated that aviation sulfate aerosols produced negative simulated radiative forcing through brightening low-level clouds. Thus, it is worth noting that the induced radiative forcing highly depends on the background cloud distribution which therefore represents an important uncertainty in quantifying the pattern and magnitude of the radiative forcing induced by aviation sulfate aerosols. This also depends on the background aerosol distribution and

assumptions made on the physics of aerosol-cloud interactions in contrails. Another factor to consider is the assumed particle size of aviation sulfate aerosols. As shown in Gettelman and Chen (2013), greater forcing can be induced when a smaller particle size is assumed. Thus, considering the much higher forcing of aviation sulfate aerosols in 2050, the assumed particle size can create a large spread in simulated results.

Uncertainties of these findings can be significant since the radiative forcing of aviation sulfate aerosols and soot is highly dependent on the assumptions made in the role they play in cloud nucleation processes. This is still an active area of research. Righi et al. (2013) reported a cooling effect of similar magnitude by aviation sulfate aerosols as in this study. Simulations using GATOR-GCMOM (Gas, Aerosol, Transport, Radiation, General-Circulation, Mesoscale, and Ocean Model) (Jacobson et al. , 2011, 2013) found a warming effect. The treatment of aviation soot in the parameterization can also lead to very different conclusion of their impact. The low radiative forcing of aviation BC found in this study is consistent with Chen and Gettelman (2013) which shows that a larger forcing could be simulated when a higher heterogeneous ice nucleation efficiency for BC aerosols is assumed. Furthermore, uncertainties can also result from the assumed particle size of aviation aerosols which is reported in Chen and Gettelman (2013).

In summary, CAM simulations indicate that future projected increases in aviation emissions may lead to large increases in contrail radiative forcing and decreases in forcing due to aviation aerosol cooling. There is warming from contrail cirrus, but significant aviation aerosol effects from sulfate result in net cooling, increasing in the future. BC does not have a significant impact on RF. The cooling arises from aerosol effects on LWP well below cruise altitude. These effects have been seen in previous studies with CAM5 (Gettelman and Chen, 2013) and other models (Righi et al., 2013).

However, net positive radiative forcing remains over land, because aerosols are most effective in creating a forcing by brightening clouds over oceans. There are regional effects up to 1–2 $\text{W m}^{-2}$ over high traffic regions of the Northern Hemisphere. Largest increases in effects between 2006 and 2050 are projected in East Asia where aviation emissions have the highest regional projected increase. Note that ambient aerosols are decreasing in the future, so effects of aviation aerosols may become more pronounced. Our results indicated minimal sensitivity in contrail cirrus radiative forcing due to fuel consumption (SC1 v. BL). On the other hand, due to the large effects of aviation sulfate aerosol, there is a strong sensitivity to alternative fuels that reduce/eliminate sulfate emissions (SC2). Finally, there is no significant sensitivity to small (5 %) aviation water vapor perturbations.

To put aviation impact in context, here we provide an estimate for the radiative forcing of aviation $CO_2$. Based on the 2005 emission level, Lee et al. (2009) reported radiative forcing of $28\,\text{mW m}^{-2}$. In 2050 under the BL scenario, the fuel

consumption is projected to increase by a factor of 5. Therefore, aviation $CO_2$ is projected to induce a positive radiative forcing of $140\,\mathrm{mW\,m^{-2}}$. This will overwhelm the net effect of contrail cirrus and aviation aerosols reported in this study and result in a net positive radiative forcing globally.

The climate response to the radiative forcing induced by aviation emissions is beyond the scope of this study. This will be addressed by employing fully coupled simulations by the Community Earth System Model (CESM) and the results will be presented in another paper.

*Acknowledgements.* This work was funded by the FAA's ATAC Program under award 10-1110-NCAR. Computing resources were provided by the Climate Simulation Laboratory at National Center for Atmospheric Research (NCAR) Computational and Information Systems Laboratory. NCAR is sponsored by the US National Science Foundation. The authors thank Brian Medeiros for his review and Rangasayi Halthore of the FAA for his support.

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

**Table 1.** Various aviation emission scenarios based on Barrett et al. (2010).

| Year | Name | Description |
|---|---|---|
| 2006 | AIR | aviation emissions based on observed data in 2006 |
| 2016, 2026, 2036, 2050 | BL | Baseline projected aviation emissions in 2016, 2026, 2036, 2050 |
| 2016, 2026, 2036, 2050 | SC1 | Scenario 1, lower fuel use assuming 2 % efficiency improvement per year |
| 2016, 2026, 2036, 2050 | SC2 | Scenario 2, SC1 with no sulfate and 50 % BC emissions |
| 2016, 2026, 2036, 2050 | SC3 | Scenario 3, SC2 with 5 % increase in emitted water |

**Table 2.** Vertices of eastern North America, central Europe, and East Asia denoted by blue boxes in Fig. 6.

| Eastern North America | Central Europe | East Asia |
|---|---|---|
| $(95^\circ$ W, $45^\circ$ N) | $(5^\circ$ W, $58^\circ$ N) | $(100^\circ$ E, $43^\circ$ N) |
| $(58^\circ$ W, $45^\circ$ N) | $(23^\circ$ E, $58^\circ$ N) | $(150^\circ$ E, $43^\circ$ N) |
| $(95^\circ$ W, $26^\circ$ N) | $(5^\circ$ W, $45^\circ$ N) | $(92^\circ$ E, $11^\circ$ S) |
| $(68^\circ$ W, $26^\circ$ N) | $(23^\circ$ E, $58^\circ$ N) | $(115^\circ$ E, $11^\circ$ S) |

**Table 3.** Radiative forcing $(\mathrm{mW\,m^{-2}})$ due to aviation $H_2O$ emissions. The uncertainties are based on two standard deviations of the four-member ensemble.

| meteorology | emission scenario | global | N. America | C. Europe | E. Asia |
|---|---|---|---|---|---|
| present | 2006AIR | 12±4 | 195±30 | 483±69 | 41±8 |
| 2050RCP4.5 | 2050BL | 87±6 | 798±152 | 1682±535 | 272±39 |
| 2050RCP4.5 | 2050SC1 | 76±7 | 724±136 | 1568±489 | 231±36 |
| 2050RCP4.5 | 2050SC2 | 76±7 | 724±136 | 1568±489 | 231±36 |
| 2050RCP4.5 | 2050SC3 | 80 | 681 | 1952 | 213 |
| 2050RCP8.5 | 2050BL | 83±3 | 852±92 | 1558±226 | 248±25 |
| 2050RCP8.5 | 2050SC1 | 73±4 | 777±96 | 1442±235 | 211±24 |
| 2050RCP8.5 | 2050SC2 | 73±4 | 777±96 | 1442±235 | 211±24 |
| 2050RCP8.5 | 2050SC3 | 75 | 865 | 1597 | 221 |

**Table 4.** Radiative forcing $(\mathrm{mW\,m^{-2}})$ due to aviation $H_2O + BC + SO_4$ emissions. Note that there is no aviation sulfate aerosols in SC2 and SC3. The uncertainties are based on two standard deviations of the four-member ensemble.

| meteorology | emission scenario | global | N. America | C. Europe | E. Asia |
|---|---|---|---|---|---|
| present | 2006AIR | −22±11 | 117±34 | 407±64 | 15±9 |
| 2050RCP4.5 | 2050BL | −71±25 | 491±200 | 1301±508 | 13±87 |
| 2050RCP4.5 | 2050SC1 | −28±18 | 514±169 | 1284±451 | 57±63 |
| 2050RCP4.5 | 2050SC2 | 73±5 | 754±103 | 1483±456 | 173±38 |
| 2050RCP4.5 | 2050SC3 | 76 | 728 | 1824 | 160 |
| 2050RCP8.5 | 2050BL | −77±21 | 363±125 | 868±408 | 44±73 |
| 2050RCP8.5 | 2050SC1 | −32±14 | 417±116 | 920±373 | 80±60 |
| 2050RCP8.5 | 2050SC2 | 69±9 | 804±80 | 1320±346 | 151±32 |
| 2050RCP8.5 | 2050SC3 | 75 | 871 | 1601 | 149 |

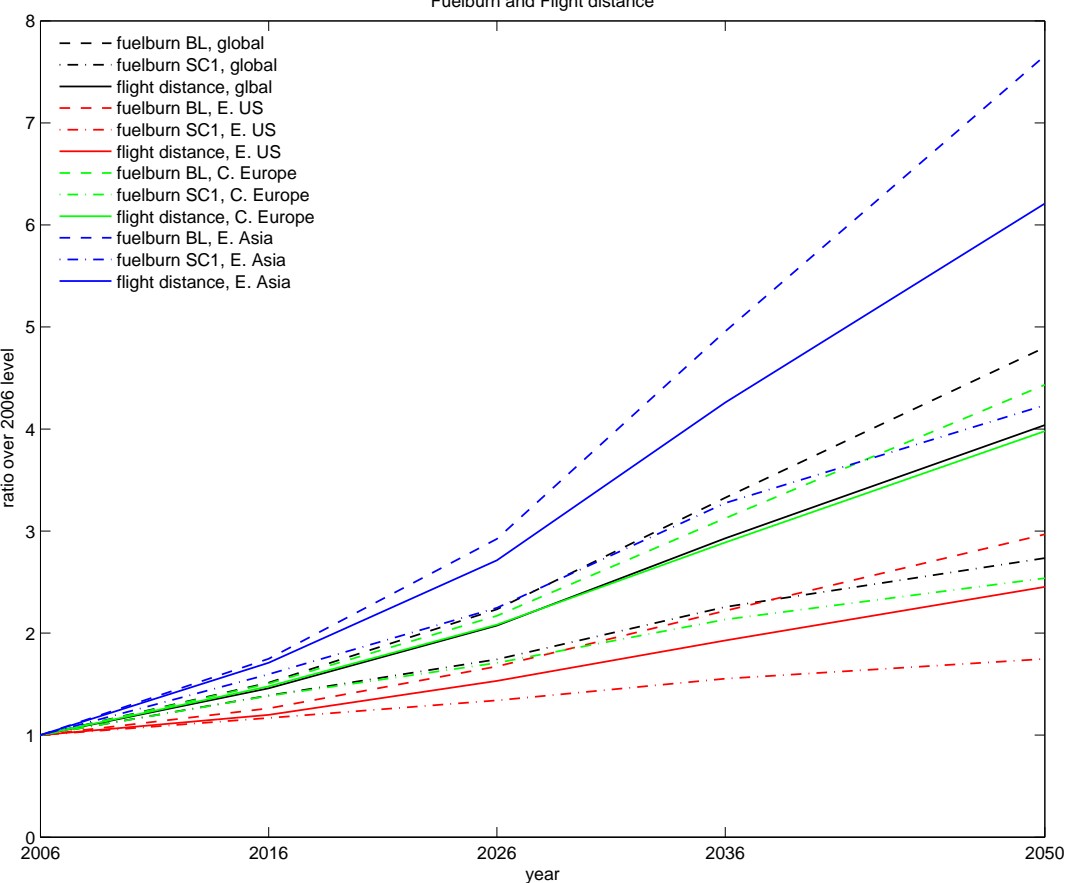

**Fig. 1.** Global and regional flight distance and fuelburn under two future scenarios, Baseline and Scenario 1, over the 2006 level. The regions over eastern US, central Europe and East Asia are depicted by the blue boxes in Fig. 6.

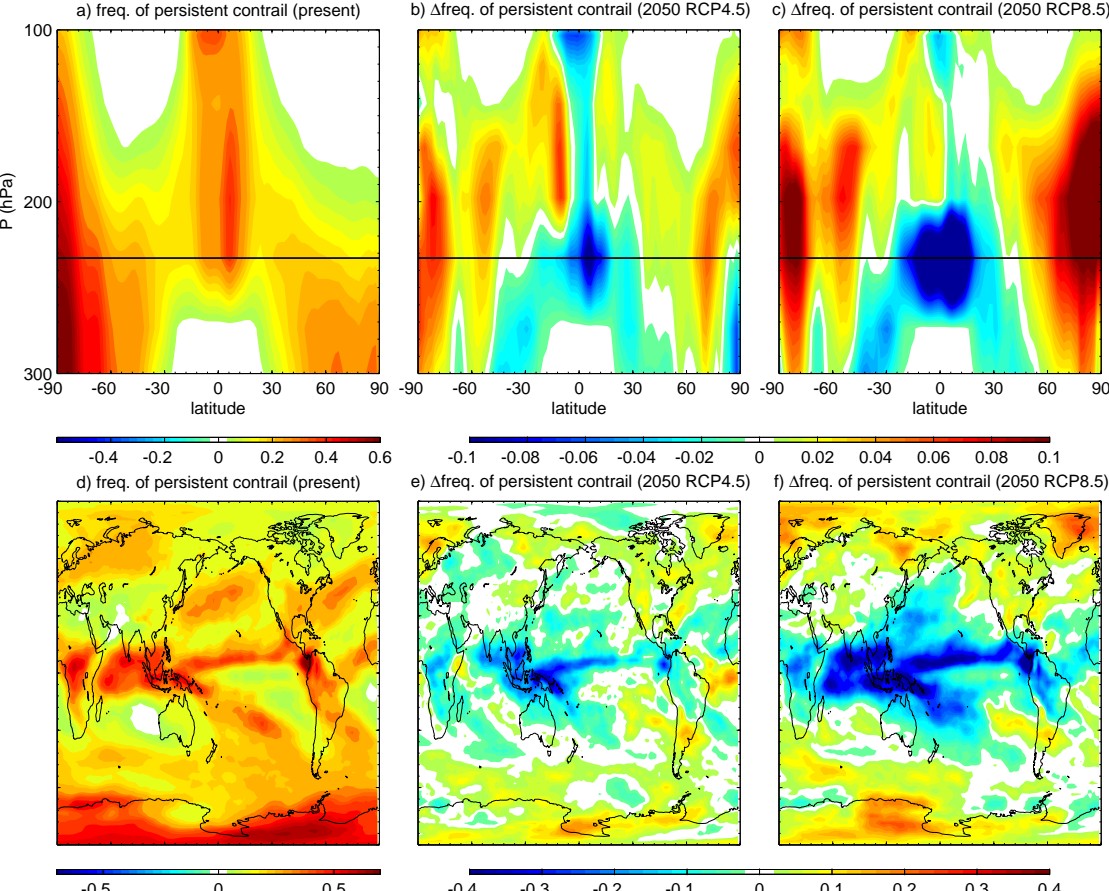

**Fig. 2.** Frequency of persistent contrail based on present and future (RCP4.5 in 2050 and RCP8.5 in 2050) meteorologies: **(a)** zonal average between 100 and 300 hPa, **(b)** zonal difference between 2050 of RCP4.5 and present, **(c)** zonal difference between 2050 of RCP8.5 and present, **(d)** the present condition at $P = 232\,\text{hPa}$, **(e)** difference between 2050 of RCP4.5 and present at $P = 232\,\text{hPa}$, and **(f)** difference between 2050 of RCP8.5 and present.

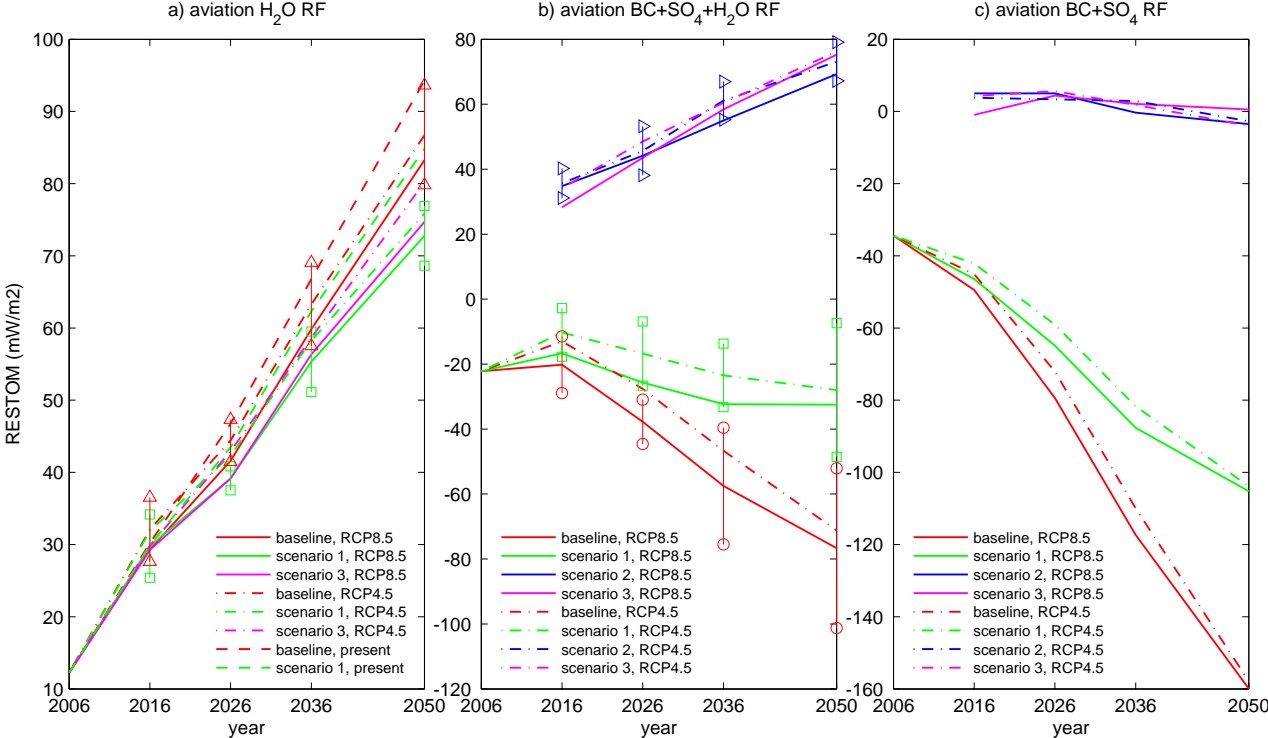

**Fig. 3.** Time series of globally averaged radiative forcing in $\mathrm{mW\,m^{-2}}$ based on various aviation emissions scenarios and background meteorologies due to: **(a)** contrail cirrus only, **(b)** contrail cirrus and aviation aerosols, and **(c)** aviation aerosols, deduced by the difference between **(a)** and **(b)**. RESTOM denotes the change in the net residual radiative flux at the top of model. Note that in **(b)** and **(c)**, there is no aviation sulfate aerosols in scenarios 2 and 3. Uncertainties plotted are based on two standard deviations of the 4-member ensemble of background meteorologies.

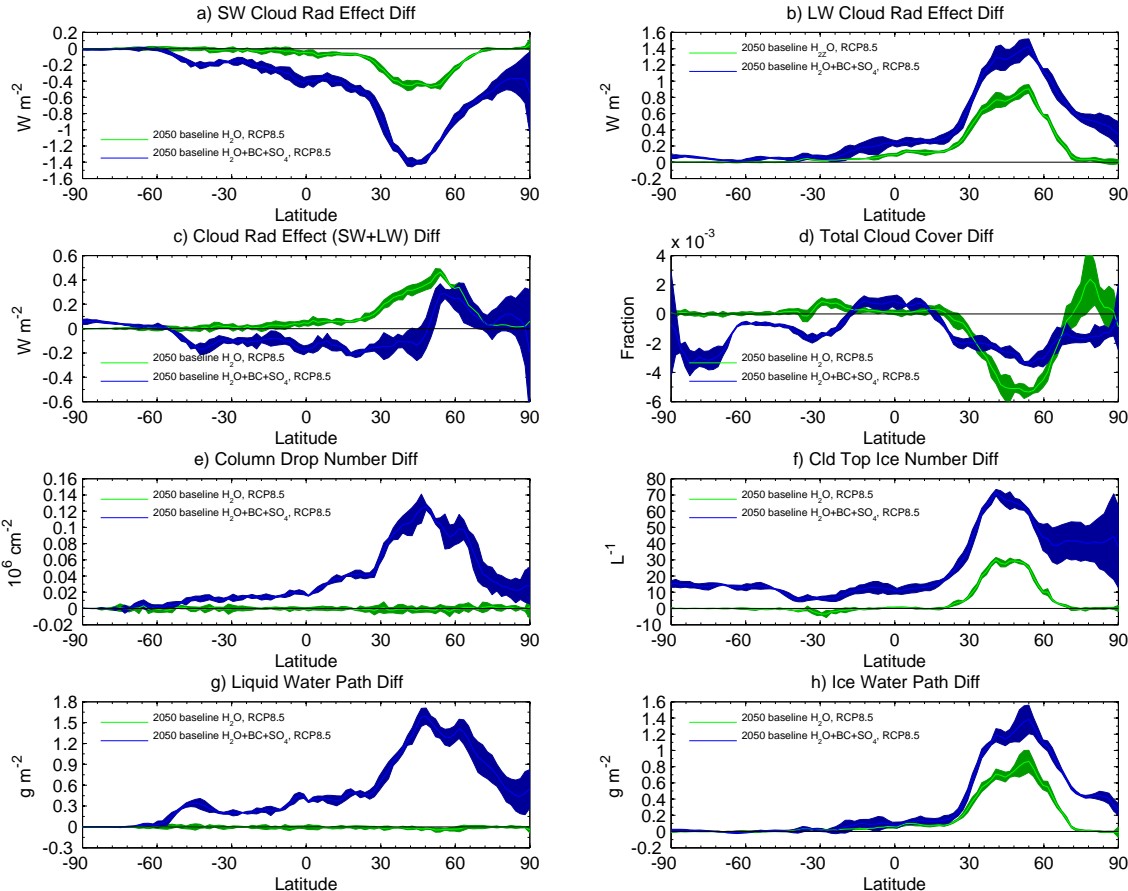

**Fig. 4.** Zonally averaged aviation impact with the green line representing water vapor emissions only of Baseline in 2050 and the blue line representing water vapor and aviation aerosols of Baseline in 2050, both coupled with the background meteorology in 2050 by RCP8.5: **(a)** shortwave cloud forcing, **(b)** longwave cloud forcing, **(c)** net cloud forcing, **(d)** total cloud difference, **(e)** column drop number difference, **(f)** cloud top ice number difference, **(g)** liquid water path difference, and **(h)** ice water difference. The spread of each curve represent the two standard deviations from the four-member ensemble of each case.

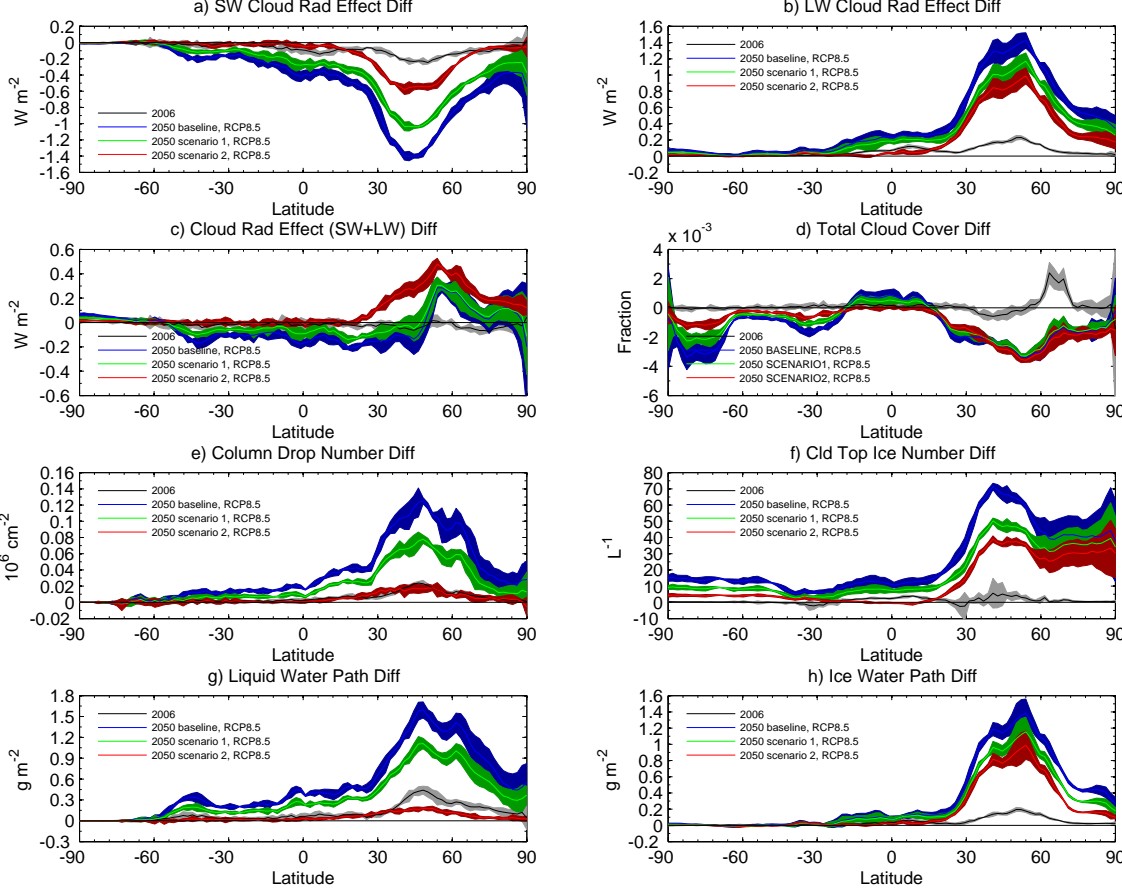

**Fig. 5.** Similar as in Fig. 4, zonally averaged aviation impact due to contrail cirrus and aviation aerosols based on emissions in 2006 and the present background meteorology, three future emissions scenarios in 2050 (Baseline, Scenarios 1 and 2) and the background meteorology in 2050 by RCP8.5.

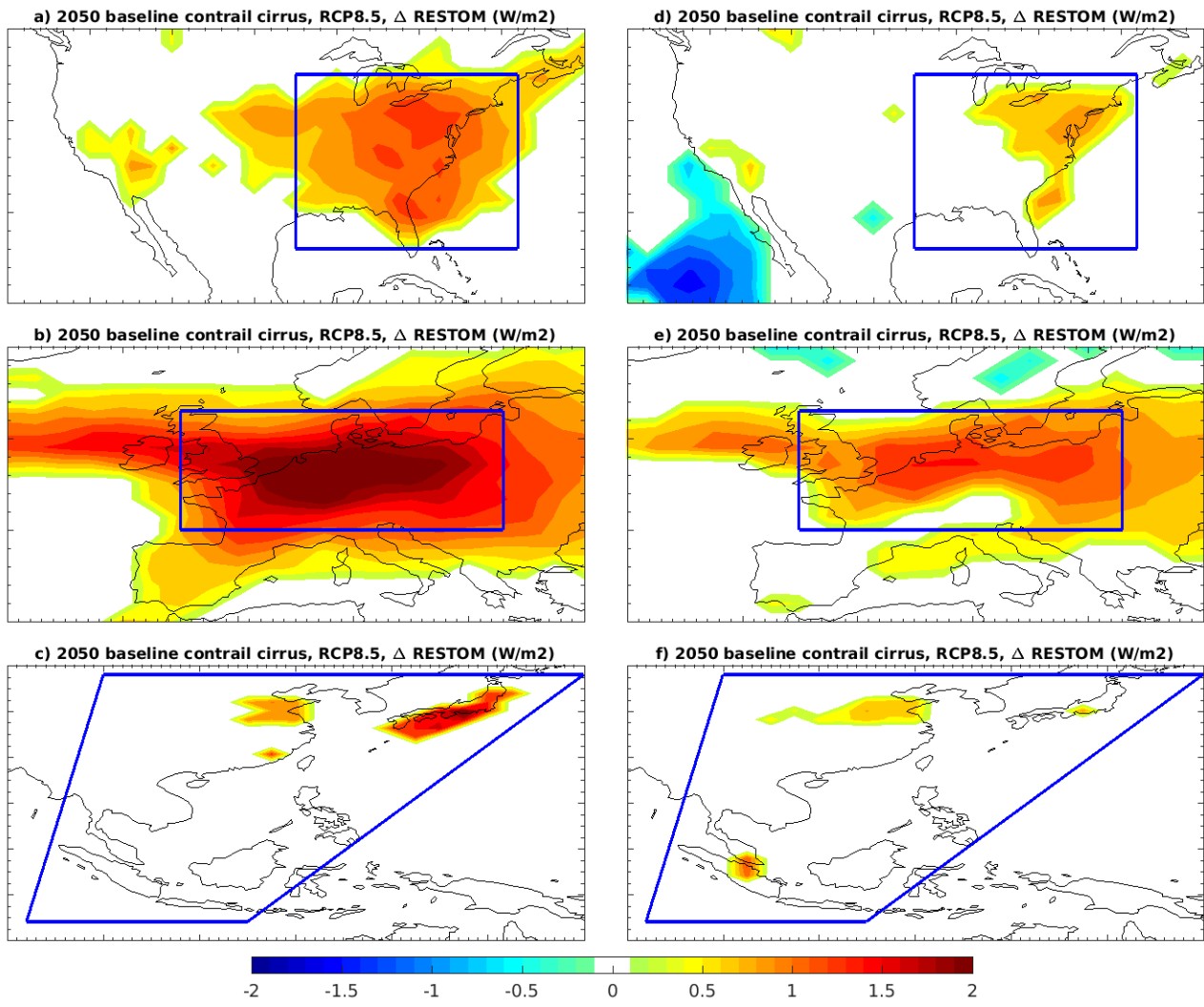

**Fig. 6.** Ensemble mean of regional radiative forcing in $\mathrm{W\,m^{-2}}$ based on the Baseline emission scenario in 2050 with RCP8.5 2050 meteorology due to contrail cirrus **(a–c)** and contrail cirrus and aviation aerosols **(d–f)**. The box averages are given in the title of each panel. Only ensemble-mean perturbations above two standard deviations of the averaged control simulations are considered in plotted.

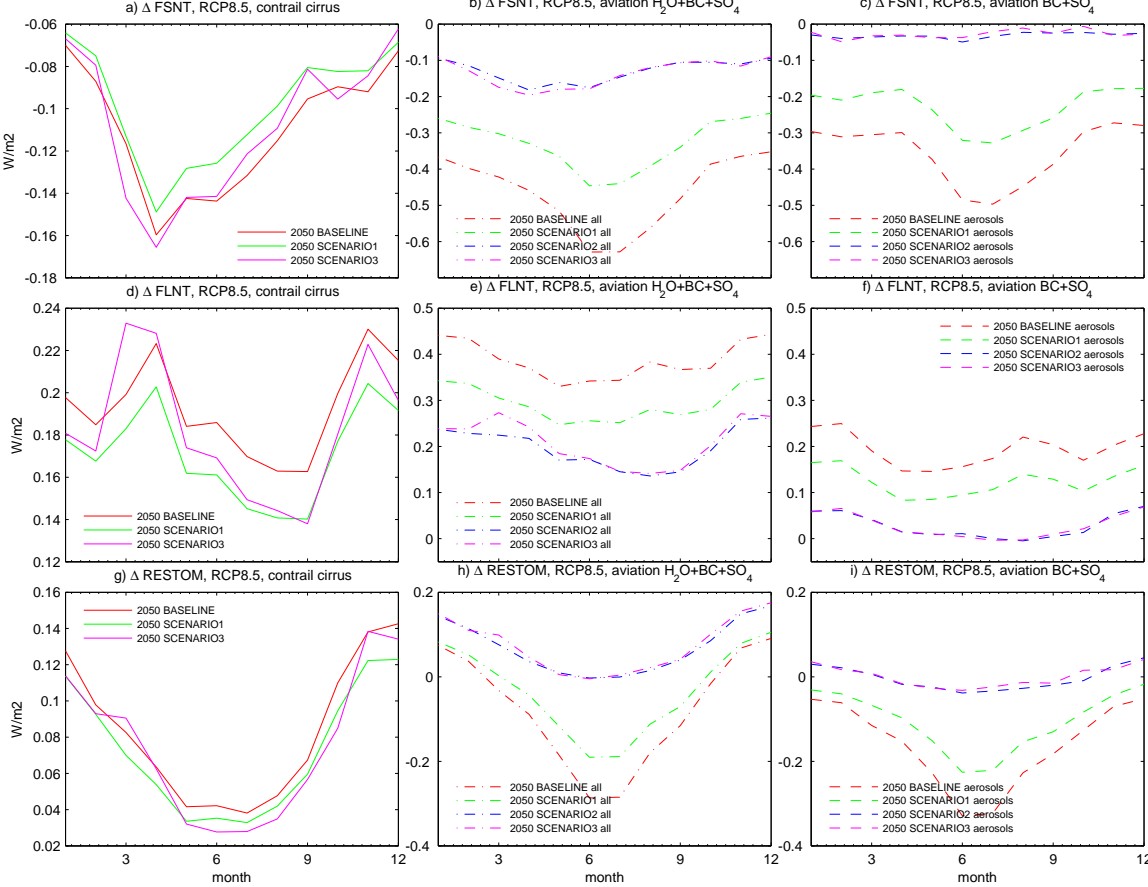

**Fig. 7.** Annual cycle of globally averaged shortwave (FSNT: **a–c**), longwave (FLNT: **d–f**) and net (RESTOM: **g–i**) radiative forcing in $\mathrm{W\,m^{-2}}$ at the top of the atmosphere due to contrail cirrus (**a, d, g**), contrail cirrus and aviation aerosols (**b, e, h**), and aviation aerosols (**c, f, i**) based on three aviation emission scenarios in 2050 with RCP8.5 2050 meteorology.

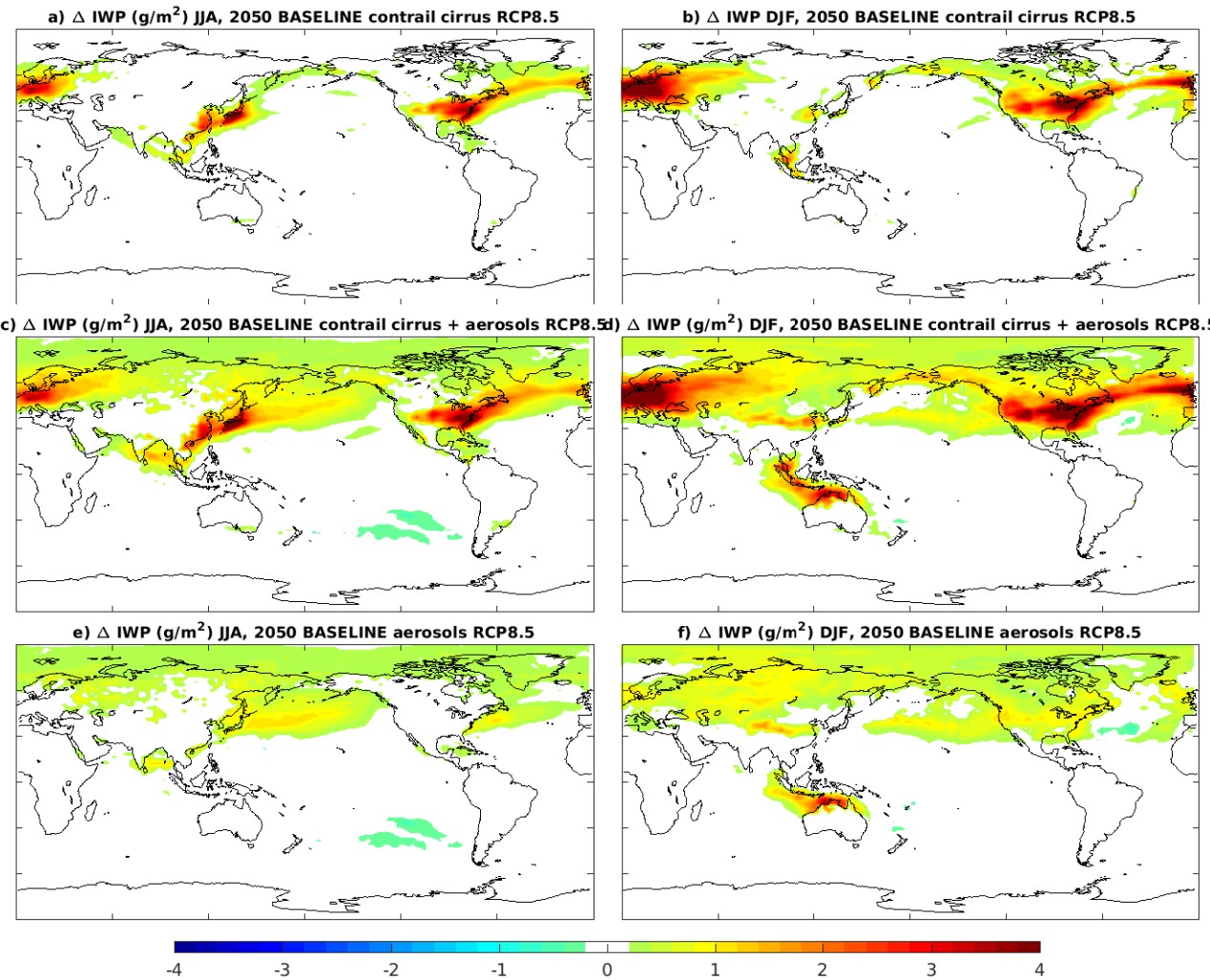

**Fig. 8.** Ensemble mean of seasonally averaged (JJA: **a, c, e**, DJF: **b, d, f**) ice water path difference in $\mathrm{g\,m^{-2}}$ due to contrail cirrus **(a, b)**, contrail cirrus and aviation aerosols **(c, d)**, and aviation aerosols **(e, f)** based on the 2050 Baseline emissions with RCP8.5 2050 meteorology. Only ensemble-mean perturbations above two standard deviations of the averaged control simulations are plotted.

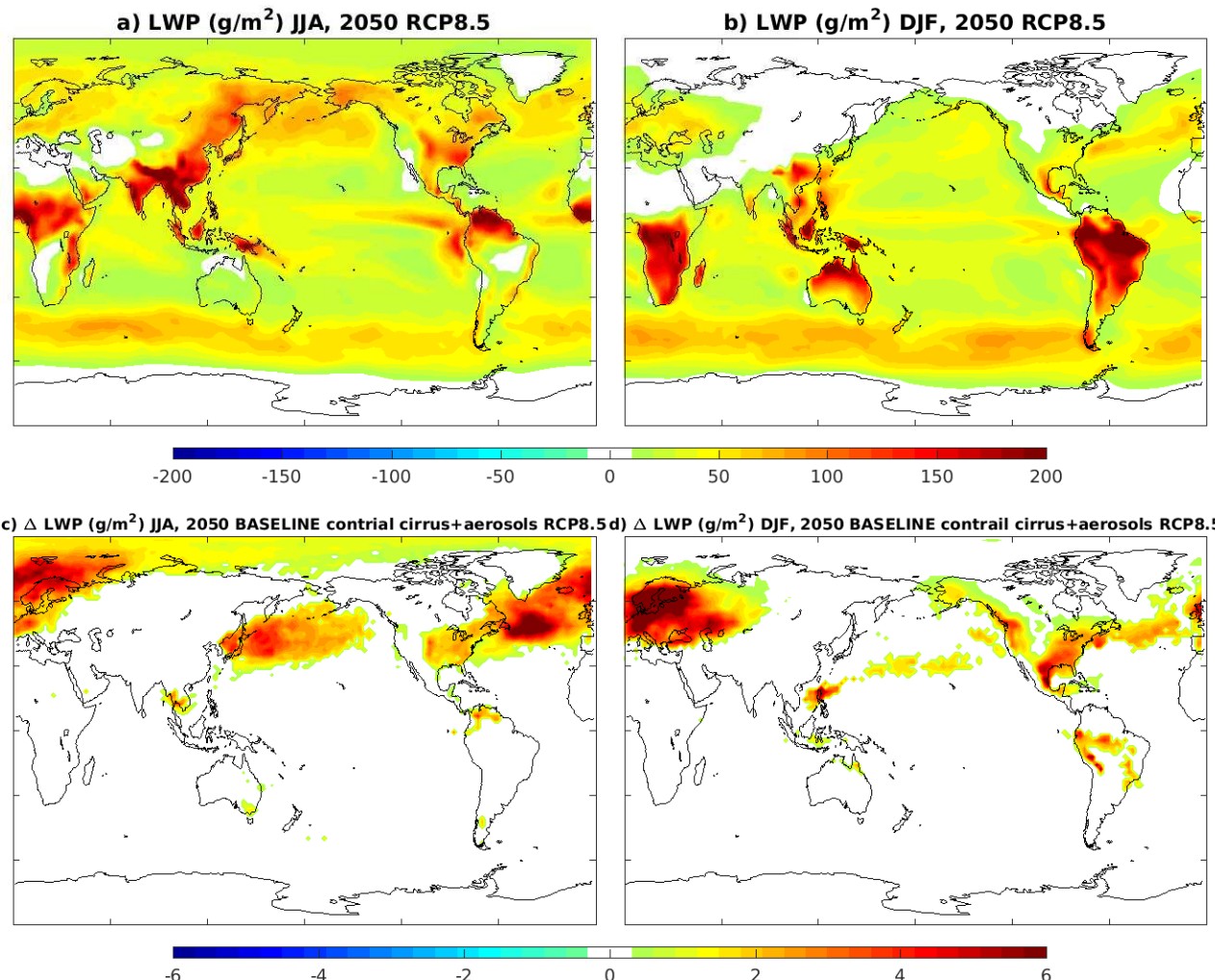

**Fig. 9.** Seasonally averaged liquid water path in $\mathrm{g\,m^{-2}}$: **(a)** JJA background field of RCP8.5 2050 meteorology, **(b)** DJF background field of RCP8.5 2050 meteorology, **(c)** JJA difference (ensemble mean) due to contrail cirrus and aviation aerosols in 2050, **(d)** DJF difference (ensemble mean) due to contrail cirrus and aviation aerosols in 2050. Only ensemble-mean perturbations above two standard deviations of the averaged control simulations are plotted.

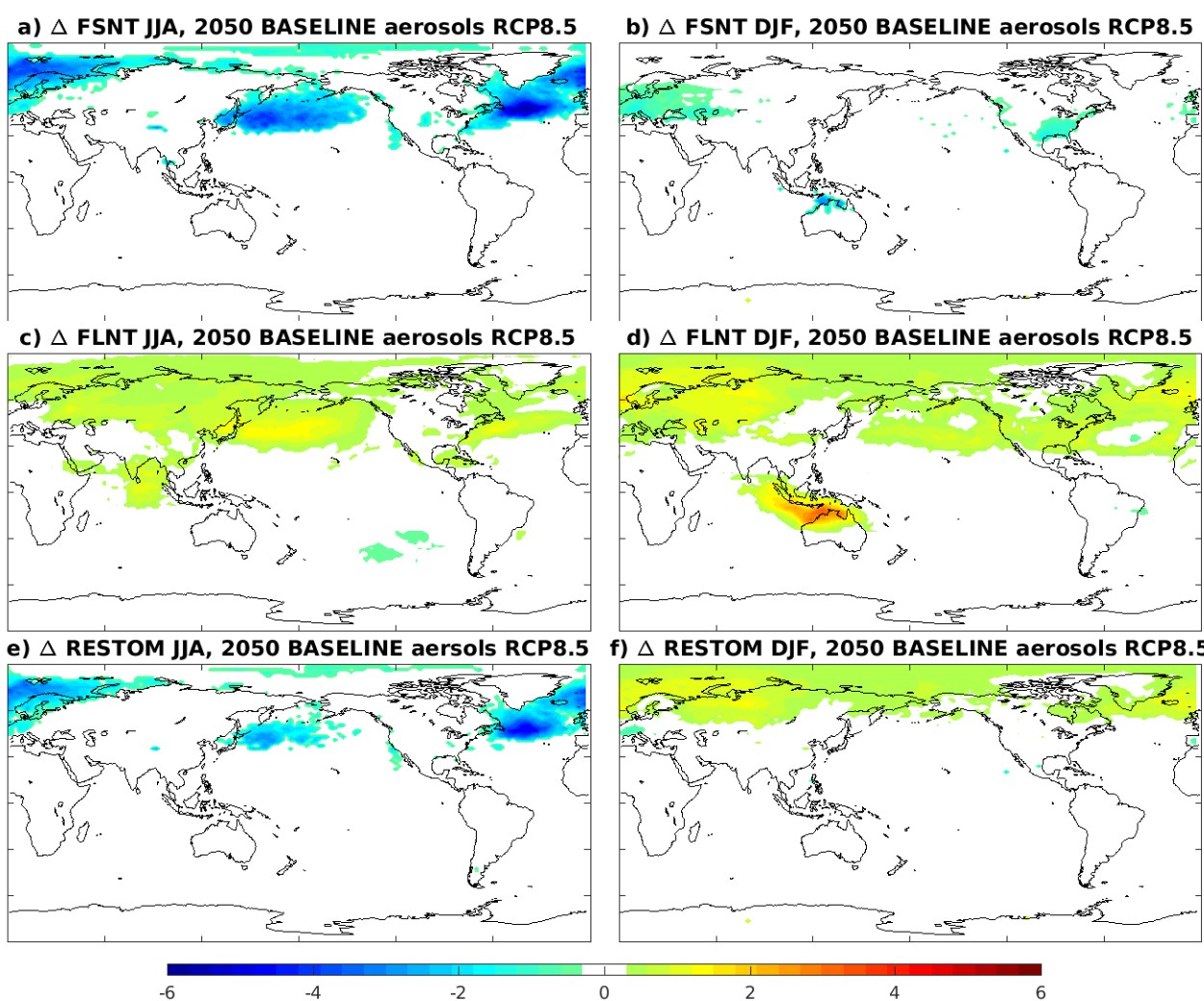

**Fig. 10.** Ensemble mean of seasonally averaged shortwave (FSNT: **a, b**), longwave (FLNT: **c, d**), and net (RESTOM: **e, f**) radiative forcing in $\mathrm{mW\,m^2}$ at the top of the atmosphere due to 2050 Baseline aviation aerosols with RCP8.5 2050 meteorology. Only perturbations above two standard deviations of the corresponding control simulation are considered in the ensemble mean.

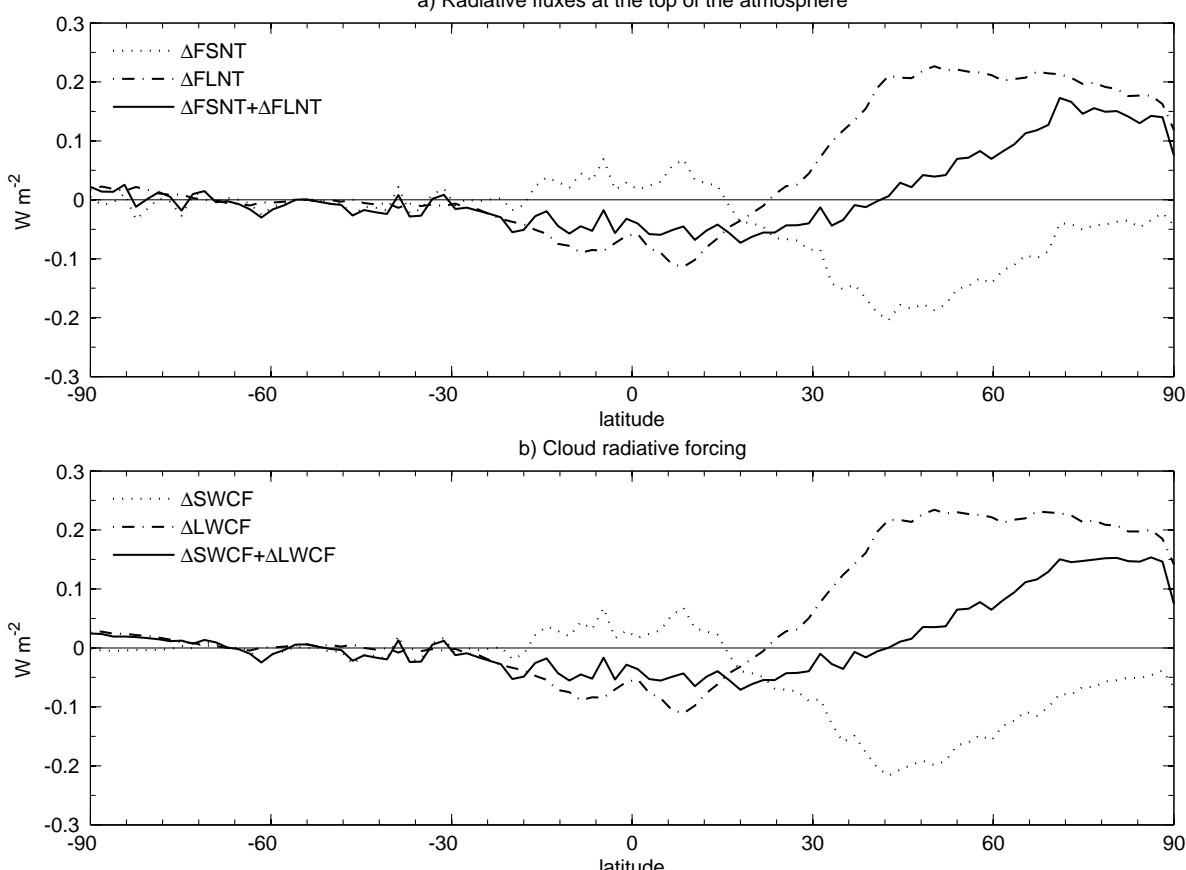

**Fig. 11.** Zonal average of radiative fluxes at the top of the atmosphere energy balance (shortwave: FSNT, longwave: FLNT) and cloud radiative forcing (shortwave: SWCF, longwave: LWCF) in $\mathrm{mW\,m^{-2}}$ due to aviation BC based on 2050BL emissions with 2050 RCP8.5 meteorology.