# Peer review of "Simulated 2050 Aviation Radiative Forcing from Contrails and Aerosols"

_Atmospheric Chemistry and Physics, 2015_

## Referee Comment (RC1) · U. Schumann (Referee) · 30 Jan 2016

The paper investigates the radiation forcing (RF) from increased air traffic in the year 2050 compared to 2006 for given scenarios using a global climate/aerosol general circulation model, in a nudged mode, with a highly approximate method to represent contrail cirrus.

The study finds an over-proportional increase of positive RF from contrails. The absolute value in 2050 stays small because the model predicts a small contrail RF also for 2006. The model finds a larger negative RF from aviation sulfate aerosols on liquid clouds (assuming that fuels still contain sulfur in 2050). They state: "As a result, the net 2050 aviation radiative forcing has a cooling effect on the planet."

The potential climate impact of aviation may be important for future climate change and any new result on this attracts attention in the aviation community and related science

and policy discussions. This requires a carefully formulated abstract and conclusions.

The results presented are straightforward extrapolations from Gettelman and Chen (GRL, 2013) who concluded: "Direct and (mostly) indirect effects on liquid clouds from SO4 of –46 mW m-2 are larger than the warming effect due to contrail cirrus and aviation induced cloudiness (16 mW m-2)." So, the new study differs only by using scenarios for future traffic.

The impact of traffic scenarios until about 2050 has been investigated before [Gierens et al., 1999; Marquart et al., 2003]. See also the discussions in [IPCC, 1999] in the chapters on aerosols, climate change, and technology. These studies are not cited here.

The paper does not explain why contrail RF increases by a factor of 7; see Table 3, mentioned on page 9 and the summary, without explaining the reason. The traffic increases by a factor of 4 on average and by a factor of 6 in Asia. The meteorological conditions show a warming with less contrails forming in the future. A contrail cover increase could be understood from an increase in the overall-propulsion efficiency eta [Schumann, 1996], but eta seems to be kept constant here (not clear). Higher efficiency of aviation requires more efficient propulsion. Hence eta should increase [Sausen et al., 1998]. So, what causes the factor 7?

A possible reason may be the low temperature (and possibly a cold bias) at the extratropical tropopause, possibly enhanced for the future climate. For higher and increased traffic at the tropopause more cirrus gets very cold (and the surface gets warmer) causing stronger LW contrail forcing. How do the temperatures in CAM5 compare with ERA-reanalysis results? Other possible reasons: does the atmosphere brightness temperature increase? Does the effective albedo increase? Both would increase the RF from contrails [Meerkötter et al., 1999].

The comparisons of the model results with observations and other model studies for present climate (here 2006), presented so far, are not stringent enough to allow for

Interactive
comment

extrapolation into the far future without careful discussion of consequences of model uncertainties for the results. The model strengths are overemphasized and the model weaknesses partly hidden. Parts of the study are not new, and related references not sufficiently acknowledged.

The abstract reports the simulation results as if one could trust them in quantity and sign. A newcomer would read from this paper that aircraft cause a negative RF at present and in the future. The tittle of the paper is misleading, since the paper discusses only a fraction of the important aviation effects ($CO_2$ is missing, for example). The abstract and the paper does not reflect all the uncertainties which exist in this model study.

The contrail cirrus model used does not compare well with observations. The tests shown in Chen et al (2013) all show large differences to observations.

Part of the problem comes from the highly simplified contrail model used. The method assumes that emission from aviation get spread over a grid cell (about 200 km * 200 km * 1 km) within half an hour. Thereafter they are part of normal cirrus and have the same optical and sedimentation properties. That may be "self-consistent" but is not physically correct. See the many recent contrail and contrail cirrus observations [Voigt et al., 2011; Iwabuchi et al., 2012; Bedka et al., 2013; Duda et al., 2013; Jeßberger et al., 2013; Minnis et al., 2013; Vázquez-Navarro et al., 2015] and LES [Lewellen, 2014; Unterstrasser, 2014].

Contrail cirrus is optically thicker than assumed some years ago [Marquart et al., 2003] and observations are coming back to estimates as of the 1999 IPCC [Iwabuchi et al., 2012; Kärcher and Burkhardt, 2013; Vázquez-Navarro et al., 2015].

Aircraft size or speed effects are ignored but are important [Voigt et al., 2011].

The ice particle concentration is computed independently of the soot number emissions. This is inconsistent with several observations and models [Kärcher and Yu,

2009].

The model underestimates the ice water content in 30-min old contrails [Schumann et al., 2015], possibly by 1 to 2 orders of magnitude.

The diurnal cycle of cirrus properties in the North Atlantic, discussed shortly in Chen and Gettelman (2913) and their response to a reviewer remark, is more than an order of magnitude smaller than observed [Graf et al., 2012].

There are further studies on line-shaped contrails not cited here, partially giving far larger RF [Kärcher et al., 2010; De Leon et al., 2012].

The model approach does not include heterogeneous ice nucleation effects from soot, possibly being preprocessed in contrails [Zhou and Penner, 2014].

Are there test results from CAM5 which can be used to assess the radiation transfer model and the background atmosphere properties for contrail cirrus in the modelled background atmosphere as shown in [Myhre et al., 2009] (and later studies based on this).

Some of the uncertainties were discussed in the preceding papers of the author team but are not reflected properly in this paper.

For example, the present paper cites the 2006 results, for present traffic, with 12 mW/m2. In the previous paper (ACP, 2013), it was stated as 13+-10 mW/m2. I now miss an assessment of the huge uncertainty range.

The paper mentions other contrail RF results, which are about 4 times larger (see also [Schumann et al., 2015]), but does not reflect these differences in the conclusions and the abstract.

The authors tend to show comparisons and say they show good agreement when the agreement is in in fact not good or at best marginal. For example, in their 2013 ACP paper they wrote: "CAM5 can simulate the mean relative humidity and reproduce the

distribution of the frequency of ice supersaturation in the upper troposphere and lower stratosphere (UTLS) (Chen et al., 2012) as observed..." If one looks to Chen et al. (2012), one notes huge differences in the panels a) and b) of Fig. 1. The text comments the figure: "Relative humidity in CAM5-SD is about 50% higher than AIRS throughout much of the UTLS." Later: "The frequency of ice supersaturation in CAM5-SD is also higher than in AIRS". Nevertheless they state: "CAM5-SD does a reasonable job..." To my opinion, this conclusion is not justified.

Chen et al. (2012) find that the model results depend strongly on vertical resolution. In the present paper this irritating fact is simply ignored.

They state in Chen et al. (2012): "CAM5-L82 is found to produce cloud fraction distributions and gradients similar to MODIS but with lower magnitude (by a factor of 3)." Chen and Gettelman (2013) give an uncertainty of factor 2.5. This uncertainty is not reflected in the new paper.

With respect to sulfate aerosols: The authors say that "Aviation aerosols emitted at cruise altitude can be transported down to near Earth's surface and thus the aerosol concentration in the lower troposphere can be substantially increased in remote regions."

I wonder whether any increases of sulfate aerosol from aviation has been observed or is observable at all. How does this increase compare with changes in aerosol concentrations from other sources (natural and shipping etc.)? There is no observational constraint to test the model results in this respect.

Hence, the aerosol part is highly speculative and this should be admitted.

The amount of aerosol arriving in low-level clouds depends strongly on the modelling of wet scavenging and precipitation reaching the ground. This is clearly discussed in the paper by Liu et al. (GMD, 2012), on which this study is based. But the many uncertainties which were discussed by Liu et al. are not taken into account here.

It would be interesting to see a parameter study on wet scavenging parameters and show how they impact the aviation effects. The scavenging of aviation aerosol is special because of the high emission altitudes, often far above liquid or mixed-phased clouds.

Gettelman and Chen (GRL, 2013) write: "The $-46$ mW m-2 represents about 3% of the $-1600$ mW m-2 total anthropogenic SW liquid cloud indirect effects in CAM5 [Gettelman et al., 2012]". In view of recent integral climate change arguments [Stevens, 2015], the total may be a bit high and this may apply to the computed aviation effects as well.

Then, why do you insist on just 0.1% BC activation. The evidence for this specific value from airborne observations of aviation soot is zero. Why should aviation soot have any similarity to biomass burning soot? How can you exclude a few percent?

There is little observational evidence on which you can base this quantitative assumption, from which far reaching conclusions are derived.

Another parameter of importance is the lifetime of aviation soot emissions in the atmosphere at cruise levels. They get emitted at high altitudes and get scavenged slowly just because their ice nucleation efficiency is low. The long lifetime may cause small but long-lasting effects and hence balance the low nucleation effects on cirrus partly. This may increase their importance.

In conclusion, the paper needs to be revised considerably before getting acceptable: The paper should identify not only the strengths but also the major weaknesses of the model, in comparison to existing studies, acknowledge previous work, explain results physically, and formulate abstract and conclusions such that the reader is aware that the results are of qualitative nature and not quantitatively reliable.

Additional references

Bedka, S. T., P. Minnis, D. P. Duda, T. L. Chee, and R. Palikonda (2013), Properties of linear contrails in the Northern Hemisphere derived from 2006 Aqua MODIS observa-

tions, Geophys. Res. Lett., 40, 772-777, doi:10.1029/2012GL054363.

De Leon, R. R., M. Krämer, D. S. Lee, and J. C. Thelen (2012), Sensitivity of radiative properties of persistent contrails to the ice water path, Atmos. Chem. Phys., 12 7893-7901.

Duda, D. P., P. Minnis, K. Khlopenkov, T. L. Chee, and R. Boeke (2013), Estimation of 2006 Northern Hemisphere contrail coverage using MODIS data, Geophys. Res. Lett., 40, doi: 10.1002/grl.50097.

Gierens, K., R. Sausen, and U. Schumann (1999), A diagnostic study of the global distribution of contrails, Part II: Future air traffic scenarios, Theor. Appl. Climat., 63, 1 - 9.

Graf, K., U. Schumann, H. Mannstein, and B. Mayer (2012), Aviation induced diurnal North Atlantic cirrus cover cycle, Geophys. Res. Lett., 39, L16804, doi: 10.1029/2012GL052590.

IPCC (1999), Aviation and the Global Atmosphere, 373 pp., Cambridge Univ. Press, Cambridge, UK.

Iwabuchi, H., P. Yang, K. N. Liou, and P. Minnis (2012), Physical and optical properties of persistent contrails: Climatology and interpretation, J. Geophys. Res., 117, D06215, doi:10.1029/2011JD017020.

Jeßberger, P., C. Voigt, U. Schumann, I. Sölch, H. Schlager, S. Kaufmann, A. Petzold, D. Schäuble, and J.-F. Gayet (2013), Aircraft type influence on contrail properties, Atmos. Chem. Phys., 13, 11965-11984, DOI: 10.5194/acp-13-11965-2013.

Kärcher, B., and F. Yu (2009), Role of aircraft soot emissions in contrail formation, Geophys. Res. Lett., 36, L01804, doi:10.1029/2008GL036649.

Kärcher, B., U. Burkhardt, M. Ponater, and C. Frömming (2010), Importance of representing optical depth variability for estimates of global line-shaped contrail radiative

forcing, PNAS, 19181-19184, doi:10.1073/pnas.1005555107.

Kärcher, B., and U. Burkhardt (2013), Effects of optical depth variability on contrail radiative forcing, Q. J. R. Meteorol. Soc., 139, 1658-1664, DOI:10.1002/qj.2053.

Lewellen, D. C. (2014), Persistent contrails and contrail cirrus. Part II: Full lifetime behavior, J. Atmos. Sci., 71, 4420-4438, DOI: 10.1175/JAS-D-13-0317.1.

Marquart, S., M. Ponater, F. Mager, and R. Sausen (2003), Future development of contrail cover, optical depth and radiative forcing: Impacts of increasing air traffic and climate change, J. Clim., 16, 2890-2904.

Meerkötter, R., U. Schumann, P. Minnis, D. R. Doelling, T. Nakajima, and Y. Tsushima (1999), Radiative forcing by contrails, Ann. Geophysicae, 17, 1080-1094, doi: 10.1007/s00585-999-1080-7.

Minnis, P., S. T. Bedka, D. P. Duda, K. M. Bedka, T. Chee, J. K. Ayers, R. Palikonda, D. A. Spangenberg, K. V. Khlopenkov, and R. Boeke (2013), Linear contrail and contrail cirrus properties determined from satellite data, Geophys. Res. Lett., 40, 3220-3226, doi: 10.1002/grl.50569.

Myhre, G., M. Kvalevag, G. Rädel, J. Cook, K. P. Shine, H. Clark, F. Karcher, K. Markowicz, A. Karda, O. Wolkenberg, Y. Balkanski, M. Ponater, P. Forster, A. Rap, and R. Rodriguez de Leon (2009), Intercomparison of radiative forcing calculations of stratospheric water vapour and contrails, Meteorol. Z., 18, 585-596, DOI 10.1127/0941-2948/2009/0411.

Sausen, R., K. Gierens, M. Ponater, and U. Schumann (1998), A diagnostic study of the global distribution of contrails. Part I: Present day climate, Theor. Appl. Climat., 61, 127 - 141, doi: 10.1007/s007040050058.

Schumann, U. (1996), On conditions for contrail formation from aircraft exhausts, Meteorol. Z., 5, 4-23.
[Figure]

Schumann, U., J. E. Penner, Y. Chen, C. Zhou, and K. Graf (2015), Dehydration effects from contrails in a coupled contrail-climate model, Atmos. Chem. Phys., 15, 11179-11199, doi:10.5194/acp-15-11179-2015.

Stevens, B. (2015), Rethinking the lower bound on aerosol radiative forcing, J. Clim., 28, 4794-4819, DOI: 10.1175/JCLI-D-14-00656.1.

Unterstrasser, S. (2014), Large eddy simulation study of contrail microphysics and geometry during the vortex phase and consequences on contrail-to-cirrus transition, J. Geophys. Res. Atmos., 119, 7537-7555, doi:10.1002/2013JD021418.

Vázquez-Navarro, M., H. Mannstein, and S. Kox (2015), Contrail life cycle and properties from 1 year of MSG/SEVIRI rapid-scan images, Atmos. Chem. Phys., 15, 8739-8749, doi:10.5194/acp-15-8739-2015.

Voigt, C., U. Schumann, P. Jessberger, T. Jurkat, A. Petzold, J.-F. Gayet, M. Krämer, T. Thornberry, and D. W. Fahey (2011), Extinction and optical depth of contrails, Geophys. Res. Lett., 38, L11806 doi:10.1029/2011GL047189.

Zhou, C., and J. E. Penner (2014), Aircraft soot indirect effect on large-scale cirrus clouds: Is the indirect forcing by aircraft soot positive or negative?, J. Geophys. Res., DOI: 10.1002/2014JD021914.
* * *

---

## Referee Comment (RC2) · Anonymous Referee #2 · 25 Feb 2016

This paper describes the radiative forcing from aviation for 2050, differentiating between contrail cirrus and aerosol effects. Different scenarios are used prescribing the amount of air traffic and changes in emissions due to increases in engine efficiency or alternative fuels. Aircraft emissions are found to have the largest impact due to sulfuric acid aerosols changing the liquid water path of clouds and therefore causing radiative cooling in particular over the oceans. If sulfur emissions are cut then nearly all the radiative forcing is due to contrail cirrus.

The paper is an extension of the authors' work on simulating the impact of contrail cirrus and aviation aerosol emissions on climate differing mainly in the inventories used. The paper is interesting and worth publication after major revisions. In particular, more information about the simulation and the parameterizations of the aviation effects needs to be given, significance tests need to be performed and the decrease in cloud coverage when simulating only the impact of contrail cirrus needs to be explained. More

critical discussion of the uncertainty of the results and of the single components would be important. Do the authors judge the uncertainty of the aerosol effect to be of the same size as that one due to contrail cirrus?

Comments:

1. You distinguish between H2O effects when talking about contrail cirrus and BC and SO4 effects when showing aerosol effects, neglecting the fact that contrail cirrus ice crystals form to a large degree on aviation aerosols and that contrail cirrus properties are dependent on the aviation aerosol emissions. You do not discuss this effect which likely means that you do not simulate the impact of soot emissions on contrail cirrus. Furthermore, it is not clear to me whether you take into account that aerosols that form contrail ice crystals may not be available for cloud indirect effects. In areas that are frequently ice supersaturated this could have a large impact on the results.

2. Throughout the paper you compare indirect aviation aerosol effects with contrail cirrus but do not discuss whether you believe the level of confidence connected with the two estimates to be similar. The uncertainty should not only consider the variability that you find within your simulations (and that you probably plot in figure 3) but also the assumptions that you make within the model such as nucleation thresholds and fraction of efficient ice nuclei. In the caption of figure 3 you should explain what kind of uncertainty you are plotting (vertical bars).

3. You seem not to adapt the Schmidt-Appleman criterion to the aviation scenarios. It is well known that a change in water vapor emissions, such as prescribed in scenario 3, or a change in fuel efficiency, that is connected with a change in propulsion efficiency, mainly impact contrail cirrus due to the change in the Schmidt-Appleman criterion and not due to a change in the water emissions. If you do adapt the criterion when prescribing different emissions, then please clearly state this.

4. When discussing the spread of contrail cirrus radiative forcing for different aviation scenarios it should be mentioned that the spread is underestimated due to the fact that

the effect of aviation aerosol emissions on contrail cirrus properties and changes in the Schmidt-Appleman criterion due to changes in water emissions and fuel efficiency are (probably) neglected (see above).

5. Could you please comment on the size of the reduction of aerosol radiative forcing between the BL case and Scenario 1 given that fuel consumption was reduced by only 2%. How can we understand this large effect?

6. In figures 4 and 5 you do not discuss your results regarding changes in total cloud cover. I do not understand why total cloud cover (fig 4d) is decreased in the northern mid latitudes when looking at 'H2O-effects' only. Shouldn't we see here the increase in total cloud cover due to contrail cirrus? Likewise in figure 5d for scenario 2 we see a reduction in total cloud cover even though in this scenario the aerosol effects are small. This needs to be analyzed and discussed in more detail.

7. Most of the figures show differences between two simulations but no significance testing is applied. This limits the information content of the figures significantly. Just to give an example, can we really expect the by far largest change in ice water path due to the indirect effect of aviation aerosols in 2050 over northern Australia during northern winter or is this signal maybe not significant (fig. 9f)? OR Do we really expect a cooling over most of Southern America resulting from changes in cloudiness in 2050 (fig. 6d)?

8. The initialization of contrails with a diameter of 10 mu is very extreme. Schroder et al. found 10-11 (7-8) mu as an effective (mass) diameter for contrails older 30 minutes. The mass based diameter for young contrails is 2 mu. Please comment on the impact on contrail cirrus ice crystal numbers and optical properties.

9. The amount of sulfur in aviation fuel varies strongly. Could you please state how much sulfur is emitted when using the base fuel.

10. When you compare the results of different simulations regarding contrail cirrus radiative forcing you should mention the base year for which contrail cirrus RF was

estimated. Part of the spread can be explained by the fact that inventories for the year 2002 and 2006 were used.

---

## Referee Comment (RC3) · Anonymous Referee #3 · 25 Feb 2016

General comments:

The article reports on global model studies of the radiative effects of aviation emissions in the time period between 2006 and 2050. The effects of contrails and aviation-induced aerosol particles are quantified. The authors discuss the individual effects of contrails and aviation-induced sulfate and black carbon aerosol in different future aviation scenarios. Since aviation shows very high growth rates it is of particularly importance to study its possible future climate effects. In this context it is quite valuable to consider the possible effects of aerosols from aviation on low-level clouds since this process has been neglected in most previous studies on the aviation impact. For these reasons, the article is of considerably high relevance for climate research, aviation industry, and environmental policy.

Since the focus of the article is the analysis of atmospheric perturbations due to specific anthropogenic emissions, including an analysis of the underlying physical and chemi-

cal process, the paper is well suited for publication in ACP. In most parts, the paper is very well written and of good technical quality. The modelling concept and the results obtained are clearly presented. Relevant literature is referenced thoroughly. Unfortunately, in-depth analysis and discussions of uncertainties are missing in some parts of the article. In addition, the number of figures appears to be somewhat unbalanced with the length of the text. I recommend publication after the following comments have been addressed by the authors.

Major comment:

As mentioned above the results of the article might be of particularly high relevance for climate research, environmental policy, and possibly also aviation industry. Decisions on future regulations of aviation emissions might be influenced by the results presented. Hence the major conclusions need to be drawn very carefully and, as a basis for robust decisions, uncertainties need to be discussed. It is of high value that the authors thoroughly consider uncertainties due to future projections of aviation emissions and background meteorology. Unfortunately, uncertainties of the major processes driving the analysed aviation effects are discussed only sparsely. More in-depth discussions would clearly improve the quality of the article. In more detail:

i) The authors mention that contrails are treated as background clouds after being initialized but refrain from discussing the possible consequences of this strong simplification. This issue could be addressed more carefully.

ii) The authors assume a comparatively low freezing fraction of aviation-induced BC typical for biomass burning smoke. A discussion on the sensitivity of the results to changes in this assumption is unfortunately missing and uncertainties due to assumptions on the freezing efficiencies of background aerosols are not discussed. More in-depth information would be helpful here. Credit should be given to the study by Zhou and Penner (JGR, 2014) who suggest large effects of aviation-induced BC on cirrus when pre-activation of these particles in contrails is assumed. It should be stressed

that the uncertainty of BC effects on cirrus is still high and further research is required to increase our understanding of this effect. The current version of the manuscript suggests that the effect is definitely negligible and no further investigations are required. Much more research would be necessary to draw such a strong conclusion.

iii) The simulations reveal that cloud modifications by sulfate aerosols from aviation have a large impact on climate. Gettelman and Chen (2013) showed that this effect strongly depends on the assumed size of the emitted particles. This uncertainty needs to be discussed in the present article as well, given the large indirect effect of the particles.

Minor comments:

1. The manuscript includes many figures. This appears to be somewhat unbalanced with the length of text. Some figures could be skipped or shifted into a supplement. For example, Fig. 6 is only briefly discussed. As an alternative, the conclusions drawn could also be discussed without showing the plots and Fig. 6 could be skipped. The plots probably show numerical noise in areas where effects are small. Hence if the figure is kept in the paper, a statistical analysis could help to exclude non-significant signals (this also holds for other plots shown in the paper).

2. 80-year simulations were performed for each time slice, driven by the respective 4 years of meteorological data extracted from the CESM experiment. Due to autocorrelations, some of these (comparatively short) 4-year periods might not be representative for the climate of the respective time slice. It should be discussed in more detail why this setup has been chosen, rather than, for instance, a transient simulation. The consequences of possible biases should be analysed.

3. The authors mention the finding of Penner et al. (ACP, 2009) that aviation BC could induce an indirect forcing of -161 mW/m2 by assuming aviation BC particles to be highly efficient heterogeneous ice nuclei. However, the authors do not mention that different sensitivity simulations were performed in that study leading to very different

results. This should be discussed in the article. Also the studies by Liu et al. (2009, JGR) and Zhou and Penner (2014, JGR) should be mentioned which suggest an indirect forcing of similar magnitude, with high uncertainty.

4. It should be mentioned in the manuscript that the downward transport of particles from aviation to low altitudes has also been found in the simulations by Barret et al (2010, Environmental Science and Technology).

5. The authors focus on contrails and aerosol-induced modification of natural clouds, due to the comparatively high uncertainties of these effects. However, facing the large growth rates in aviation also other effects might experience considerable increases. To put the conclusions of the paper in the right context, the manuscript could be improved by including some rough estimates at least of the expected $CO_2$ effect.

6. With regard to the assumed future scenarios, the underlying assumptions on regulations or technological developments could be discussed in some more detail. This would help the reader to evaluate the reachability of specific aims of climate impact mitigation, as considered in the scenarios. If these aspects are discussed somewhere else in the literature, corresponding citations would be sufficient.

7. Page 4, lines 22-23, 'BC and sulfate aerosols are internally mixed in the modes ...': It should be specified which modes are meant here, the aged modes only or the primary carbon mode as well.

8. The simulations are based on RCPs 4.5 and 8.5. It should be explained in the manuscript why these scenarios have been chosen and why RCPs 2.6 and 6.0 are not considered.

9. Fig. 3a: The authors should specify in the manuscript whether the 'different background meteorology' (present-day, RCPs 4.5 and 8.5) also implies corresponding differences in background emissions.

10. Page 14, 'inclusion of aviation aerosols is found to further increase IWP (Fig. 9c

and d)': The authors should discuss the plausibility of this effect. Is this an effect of BC or sulfate? Effects of sulfate aerosols on cirrus via homogeneous freezing are not very probable since this process is usually not limited by aerosol number. Other mechanisms could be diabatic effects due to aerosol-radiation interactions or effects on mixed phase clouds. Are the simulations capable to identify possible reasons?

11. Section 3.5, Aviation BC: As discussed, for instance, by Zhou and Penner (2014) the effects of aviation BC probably depend on the assumptions on ice nucleation efficiency of background ice nuclei as mineral dust or BC from non-aviation sources (see also major comment). A possible mechanism causing the positive radiative forcing of aviation BC simulated in the present study could be that the number of ice nuclei increases, resulting in more but smaller heterogeneously forms ice crystals. Sedimentation and corresponding thinning of cirrus would be reduced resulting in a net warming. The strength of this effect would, however, depend on the amount of ice nuclei in the background. This should be admitted in the paper.

---

## Author Comment (AC1) · 12 May 2016

Dear Dr. Schumann,

We have provided more description as noted. However, we also note that we did not provide all the details initially since all the information is available in numerous previous work, and we did not want to duplicate. We agree that some additional explanation is warranted and have provided it. We did not intend to hide anything about the model, but strike a balance between referencing previous work and providing important information in this study. We have enhanced the description along the lines that you have suggested, and we appreciate your careful review to help improve this manuscript.

General comment:

Much of the criticism concerns the approximation in the method. We understand that

you disagree with the approach we have taken to approximate contrails in a climate model, with an approximate treatment of contrails but an explicit treatment of climate. This paper is not about the methodology used in the model. We have made an effort to better describe the uncertainties and approximations were appropriate, but we are not going to restate the model methodology. This is contained in several earlier papers. We have tried to address these points and reflect the uncertainty in the methods and how that translates into uncertainty in the results. We have added to the discussion of uncertainty. We also added a paragraph at the end of the model description section that clearly indicates this philosophy and the significant uncertainties that arise.

Here is our response to your review on our manuscript:

The paper investigates the radiation forcing (RF) from increased air traffic in the year 2050 compared to 2006 for given scenarios using a global climate/aerosol general circulation model in a nudged mode, with highly approximate method to represent contrail cirrus.

*This method is approximate, but has been shown to be relevant and useful for climate studies.*

The study finds an over-proportional increase of positive RF from contrails. The absolute value in 2050 stays small because the model predicts a small contrail RF also for 2006. The model finds a larger negative RF from aviation sulfate aerosols on liquid clouds (assuming that fuels still contain sulfur in 2050). They state: "As a result, the net 2050 aviation radiative forcing has a cooling effect on the planet."

The potential climate impact of aviation may be important for future climate change and any new result on this attracts attention in the aviation community and related science and policy discussions. This requires a carefully formulated abstract and conclusions.

*We have tried to make sure we carefully formulate the abstract and conclusions, and think these have been improved by these comments, and those of other reviews.*

The results presented are straightforward extrapolations from Gettelman and Chen (GRL, 2013) who concluded: "Direct and (mostly) indirect effects on liquid clouds from $SO_4$ of -36 mWm$-2$ are larger than the warming effect due to contrail cirrus and aviation induced cloudiness (16 mWm$-2$). So, the new study differs only by using scenarios for future traffic.

*Yes, but this is important to show the impact, and the sensitivity to different emissions trajectories.*

The impact of traffic scenarios until about 2050 has been investigated before [Gierens et al., 1999; Marquart et al., 2003]. See also the discussions in [IPCC, 1999] in the chapters on aerosols, climate change, and technology. These studies are not cited here.

*Studies added.*

The paper does not explain why contrail RF increase by a factor of 7; see Table 3, mentioned on page 9 and the summary, without explaining the reason. The traffic increases by a factor of 4 on average and by a factor of 6 in Asia. The meteorological conditions show a warming with less contrails forming in the future. A contrail cover increase could be understood from an increase in the overall-propulsion efficiency eta [Schumann, 1999], but eta seem to be kept constant here (not clear). Higher efficiency of aviation requires more efficient propulsion. Hence eta should increase [Sausen et al., 1998]. SO, what causes the factor 7?

*We have added explanation on why contrail radiative forcing in 2050 in increased by a factor of 7 while the fuelburn is only increased by a factor of 5. Two factors should be considered for this. First, the fuelburn in East Asia is projected to increase by a factor of 7.5 under the baseline scenario. The radiative forcing in this region reflects this. Due to the region being lower in latitude than Central Europe and East US, the pronounced increase of contrail cirrus in East Asia can carry more weighting in the global average. Second, an important portion of contrail ice mass is from the uptake of the ambient water vapor. Under our parameterization, the volume of fresh contrails is a function of flight distance. As the flight distance increases, as projected in the future, the water uptake is also going to increase which will add to the total ice mass of the contrails.*

A possible reason may be the low temperature (and possibly a cold bias) at the extratropical tropopause, possibly enhanced for the future climate. For higher and increased traffic at the tropopause more cirrus gets very cold (and the surface gets warmer) causing stronger LW contrail forcing. How do the temperatures in CAM5 compare with ERA-reanalysis results? Other possible reasons: does the atmosphere brightness temperature increase? Does the effective albedo increase? Both would increase the RF contrails [Meerkötter et al., 1999].

*As noted above in the reply we have investigated this in detail. It is not due to cold bias.*

The comparisons of the model results with observations and other model studies for present climate (here 2006), presented so far, are not stringent enough to allow for extrapolation into the far future without careful discussion of consequences of model uncertainties for the results. The model strengths are overemphasized and the model weakness partly hidden. Parts of the study are not new, and related references not sufficiently acknowledged.

*The parts that are not new we have tried to reference (based on previous work). We have not tried to write a review of all previous work, but have added the suggested references.*

The abstract reports the simulation result as if one could trust them in quantity and sign. A newcomer would read from this paper that aircraft cause a negative RF at present and in the future. The title of the paper in misleading, since the paper discusses only a fraction of the important effects ($CO_2$ is missing, for example). The abstract and the paper does not reflect all the uncertainties which exist in this model study.

*The title has been changed to clarify the radiative forcing we are assessing. The abstract and conclusions have been modified to better highlight the uncertainties. We do not mean to imply the model is truth.*

The contrail cirrus model used does not compare well with observations. The tests shown in Chen et al. (2013) all show large differences to observations.

*We do think the model can reproduce important aspects of the distribution of contrails as noted in previous work. All models are different than observations. But it also reproduces important aspects of observations. So the reader can assess their level of comfort. We are focusing on the self-consistent climate representation of contrails.*

Part of the problem comes from the highly simplified contrail model used. The method assumes that emission from aviation gets spread over a grid cell (about 200 km * 200 km * 1 km) within half an hour. Thereafter they are part of normal cirrus and have the same optical and sedimentation properties. That may be "self-consistent" but is not physically correct. See the many recent contrail and contrail cirrus observations [Voigt et al., 2011; Iwabuchi et al., 2012; Bedka et al., 2013; Duda et al., 2013; Je$\beta$berger et

al., 2013; Minnis et al., 2013; Vázquez-Navarro et al., 2015] and LES [Lewellen, 2014; Unterstrasser, 2014].

*The reviewer is mistaken: contrails are not spread over the grid box in half an hour, as the contrail is given a small cloud fraction, representative of a hour hour of linear contrail with spreading to a few hundred meters, and added for all flights. This is detailed in Chen and Gettelman 2013. We believe this is physically correct, and the method has been published previously. The real issue with the method is spreading contrails in the vertical as the model layers are nearly 1 km thick in the UTLS. These uncertainties and sensitivity tests are detailed in earlier work as well.*

Contrail cirrus is optically thicker than assumed some years ago [Marquart et al., 2003] and observations are coming back to estimates of the 1999 IPCC [Iwabuchi et al., 2012; Kärcher and Burkhardt, 2013; Vázquez-Navarro et al., 2015].

*We are not making any assumptions that are based on a optical thickness. Our work has been checked against recent observations by Minnis et al., 2013.*

Aircraft size or speed effects are ignored but are important [Voigt et al. 2011].

*Aircraft size effects are taken care of in fuel burn. Yes, we are approximating details using this method, but this is necessary for climate purposes.*

The ice particle concentration is computed independently of the soot number emissions. This is inconsistent with several observations and models [Kärcher and Yu, 2009].

*This is a valid point, however there is large uncertainty in the nucleation properties of soot, and the number concentration.*

The model underestimates the ice water content in 30-min old contrails [Schumann et al., 2015], possibly by 1 to 2 orders of magnitude.

*We never mention ice water content in this paper, so we are not sure the origin of the comment. It may be about earlier work. There may be confusion about the method. As detailed in Chen et al. 2012, we are putting water vapor emissions from the aircraft into a contrail, and then also moving ambient humidity above ice supersaturation into the contrail. This yields a grid-box IWC. We use an expression from Schumann (2002) as a function of temperature based on observations of contrail IWC to adjust the cloud fraction so that the contrail 'in cloud' IWC matches observations.*

The diurnal cycle of cirrus properties in the North Atlantic, discussed shortly in Chen and Gettelman (2013) and their response to a reviewer remark, is more than an order of magnitude smaller than observed [Graf et al., 2012].

*This is not about this paper. This is a criticism of earlier work. We are not using the diurnal cycle in this work. The Graf et al. 2012 method also has uncertainties, but this is not the place to debate them.*

There are future studies on line-shaped contrails not cited here, partially giving far larger RF [Kärcher et al., 2010; De Leon et al., 2012].

*The references have been included in the manuscript.*

The model approach does not include heterogeneous ice nucleation effects from soot, possibly being preprocessed in contrails [Zhou and Penner, 2014].

*We have noted the caveats to the model treatment of BC in response to this and another reviewer. As noted above, there are large uncertainties here, and disagreements*

*between assumptions and laboratory experiments with ice nucleation. We have done
sensitivity tests with enhanced soot activation in earlier work.*

Are there test results from CAM5 which can be used to assess the radiation transfer
model and the background atmosphere properties for contrail cirrus in the modelled
background atmosphere as shown in [Myhre et al., 2009] (and later studies based on
this).

*The reviewer should consult the references for CAM5 contained in the model descrip-
tion in Section 2.1 in the paper and earlier work by Chen and Gettelman 2013. We
are not going to evaluate the climate model here. However, there have been numerous
studies on the development of the model cloud optics and radiation code. Gettelman
et al. 2010 describes in detail the ice particle optics used. This is cited in our previous
work.*

Some of the uncertainties were discussed in the preceding papers of the author team
but are reflected properly in this paper.

*We have tried to bring appropriate uncertainties forward, but will not bring all of them
forward. We cannot restate all of the uncertainties in the method, and we clearly refer-
ence the sources. Readers are free, like the reviewer, to examine those uncertainties
in previous work. We understand if the reviewer does not agree with the earlier as-
sessment, but that is not the point of this paper.*

For example, the present paper cites the 2006 results, for present traffic, with 12
mW/m2. In the previous paper (ACP, 2013), it was stated as 13±10 mW/m2. I now
miss an assessment of the huge uncertainty range.

*Added uncertainty range.*

The paper mentions other contrail RF results, which are about 4 times larger (see also [Schumann et al., 2015]), but does not reflect these differences in the conclusions and the abstract.

*Again, this relates to previous work and not to this manuscript. This is a critique of earlier papers.*

The authors tend to show comparisons and say they show good agreement when the agreement is in fact not good or at best marginal. For example, in their 2013 ACP paper they wrote: "CAM5 can simulate the mean relative humidity and reproduce the distribution of the frequency of ice supersaturation in the upper troposphere and lower stratosphere (UTLS) (Chen et al., 2012) as observed ..." If one looks to Chen et al. (2012), one notes huge differences in the panels a) and b) of Fig. 1. The text comments the figure: "Relative humidity in CAM5-SD is about 50% higher than AIRS throughout much of the UTLS. Later: "The frequency of ice supersaturation in CAM5-SD is also higher than in AIRS". Nevertheless they state: "CAM5-SD does a reasonable job ..." To my opinion, this conclusion is not justified.

*The conclusions are in previous papers. We respectfully disagree with the reviewer on this point, but it does not have relevance for this manuscript.*

Chen et al. (2012) find that the model results depend strongly on vertical resolution. In the present paper this irritating fact is simply ignored.

*The reviewer is correct. In earlier work we identified that the upper tropospheric climate of the GCM in this version is sensitive to the vertical resolution. Thus it is necessary to*

*use a GCM resolution that provides a different background state. It is not the contrail parameterization that is the real problem here. We have noted this now in the text.*

They state in Chen et al. (2012): "CAM5-L82 is found to produce cloud fraction distributions and gradients similar to MODIS but with lower magnitude (by a factor of 3)." Chen and Gettelman (2013) give an uncertainty of factor 2.5. This uncertainty is not reflected in the new paper.

*The uncertainty is reflected in the new stated error bars.*

With respect to sulfate aerosols: The authors say that "Aviation aerosols emitted at cruise altitude can be transported down to near Earth's surface and thus the aerosol concentration in the lower troposphere can be substantially increased in remote regions."

I wonder where any increases of sulfate aerosols from aviation has been observed or is observed at all. How does this increase compare with changes in aerosol concentration from other sources (natural and shipping etc.)? There is no observational constraint to test the model results in this respect.

*The increase of sulfate aerosols in the lower troposphere due to aviation has also been reported in Barrett et al. (2010b) which is now cited in the manuscript. This is a critical uncertainty and we have more carefully identified it in the discussion in detail: there is uncertainty due to two factors that need to be better quantified. First, the model has a large perturbation to sulfate in flight corridors. Second, the cloud response to aerosols needs to be carefully quantified. The latter is a subject of considerable ongoing work. We note these uncertainties specifically now, and the magnitudes of the effects do seem large, so if anything this is a high estimate of the effects.*

Hence, the aerosol part is highly speculative and this should be admitted.

*We note that this finding is highly uncertain. However, taking a state of the art climate model, and adding these aerosol distributions in the upper troposphere is justified and appropriate. The reviewer should refer to Gettelman and Chen, 2013 (GRL) for a more complete discussion. Furthermore, there is other work indicating this. We have noted that the aerosol portion needs to be addressed and think we have treated this as a model finding that needs to be considered. Abstract and conclusions have been modified.*

The amount of aerosol arriving in low-level clouds depends strongly on the modelling of wet scavenging and precipitation reaching the ground. This is clearly discussed in the paper by Liu et al. (GMD, 2012), on which this study is based. But the many uncertainties which were discussed by Liu et al. are not taken into account here.

*The same model is used. So these uncertainties are taken into account. Aviation sulfate will work like any other sulfate aerosols in clouds. Again, we are not pulling all uncertainties forward and redoing the work of Gettelman and Chen, 2013.*

It will be interesting to see a parameter study on wet scavenging parameters and show how they impact the aviation effects. The scavenging of aviation aerosols is special because of the high emission altitudes, often far above liquid or mixed-phased clouds.

*Agreed, but beyond the scope of this work.*

Gettelman and Chen (GRL, 2013) write: "The -46 mWm-2 represents about 3% of the -1600 mWm-2 total anthropogenic SW liquid cloud indirect effects in CAM5 [Gettelman

et al., 2012]". In view of recent integral climate change arguments [Stevens, 2015], the total may be a bit high and this may apply to the computed aviation effects as well.

*This is referencing earlier work, and is not relevant here. In any event, the point is to say it is 3% of the indirect effects, and if these are overstated in the model, the 3% should stand.*

Then, why do you insist on just 0.1% BC activation. The evidence of this specific value from airborne observations of aviation soot is zero. Why should aviation soot have any similarity to biomass burning soot? How can you exclude a few percent?

*This is referenced in Gettelman and Chen, 2013: the value comes from laboratory measurements of ice nucleation properties of BC. We have added caveats to the discussion of aviation BC to note that the effects are highly uncertain.*

There is little observational evidence on which you can base this quantitative assumption, from which far reaching conclusions are derived.

*See above. We have conducted earlier sensitivity tests of this work. The criticism is again criticism of earlier work.*

Another parameter of importance is the lifetime of aviation soot emissions in the atmosphere at cruise levels. They get emitted at high altitudes and get scavenged slowly just because their ice nucleation efficiency is low. The long lifetime may cause small but long-lasting effects and hence balance the low nucleation effects on cirrus partly. This may increase their importance.

*This effect is included in the model: low efficiency of ice nucleation for soot, but they will have a long lifetime in the upper troposphere due to a lack of scavenging.*

In conclusion, the paper needs to be revised considerably before getting acceptable: The paper should identify not only the strengths but also the major weaknesses of the model, in comparison to existing studies, acknowledge previous work, explain results physically, and formulate abstract and conclusions such that the reader is aware that the results are of qualitative nature and not quantitatively reliable.

*We have tried to note the caveats, but we are not going to restate the earlier work that addresses all these uncertainties. That work is clearly referenced and described. We have tried to add more caveats and a representation of uncertainties to the abstract and the conclusions.*

Regards,

Chih-Chieh Chen and Andrew Gettelman

---

## Author Comment (AC2) · 12 May 2016

Dear Reviewer,

We appreciate the suggestions and comments you made on our manuscript. We have modified our manuscript accordingly. We have added more description about our simulations and the parameterization. We have also performed significant tests to better represent regional perturbations induced by aviation emissions. We have also explained in detail why cloud cover in mid latitudes of the Northern Hemisphere may be reduced in 2050 when simulating only the impact of contrail cirrus. The text has been revised to acknowledge high uncertainties due to the treatment of aviation aerosols in the ice nucleation processes. We have also added the uncertainties of radiative forcing resulting from the background meteorology.

[Figure]

Here are the response to your comments.

1. When we simulate the aviation $H_2O$ effect, we do not consider the role aviation aerosols may play in the ice nucleation processes upon the formation of contrails. We have acknowledged this in the manuscript by citing work with such consideration, and remind readers that this can create great uncertainties in the results.

   We do address soot effects: based on the assumptions we make, they are small (Gettelman and Chen, 2013). We recognize there is large uncertainty in this and we have noted the uncertainties in previous work. The sensitivity to assumptions about the particle size and the ice nucleation efficiency is also addressed in the previous work.

   We do take into account that aerosols which form ice crystals may not participate in indirect effects: aviation aerosols may be scavenged by the clouds that form as contrail cirrus.

2. We have added discussion to address the uncertainties caused by the assumptions made for the role aviation aerosols play in the ice nucleation processes. The assumptions about nucleation thresholds and fraction of efficient ice nuclei are treated in our earlier papers (Chen and Gettelman, 2013, Gettelman and Chen, 2013). The sulfate has less uncertainty regarding efficiency, and the BC uncertainty may be high.

   We have revised the caption of Fig. 3 to explain the uncertainty plotted is due to the background meteorology.

3. We kept the propulsion efficiency constant in this study and we have revised the manuscript to reflect this.

4. We have added discussion to remind readers that different results may be obtained by changing the propulsion efficiency.

[Figure]

5. The reduction of fuel burn from BL to SC1 is almost 50% (please see Fig. 1).

6. We have added discussion on the reduction of cloud cover. For this, there are two competing factors to consider. The first is the additional cloud fraction due to the formation of contrails. The second is due to the uptake of ambient water vapor when contrails form which reduces the relative humidity. A reduction in relative humidity will lead the cloud scheme to lower the overall cloud fraction in the grid cell. The results reveal that as the flight distance in 2050 increases significantly from 2006, the second factor has become more important and thus leads to a reduction in cloud cover.

7. We have revised figures to reflect the uncertainty. Figs.6, 8, 9, 10 only consider perturbations above two standard deviations of the corresponding control simulation in the ensemble mean calculation.

8. We have added comments to address the uncertainties resulting from the assumption of the initial ice particle size for contrails. However, we are looking at contrails which are 15-30 minutes old. Our treatment is consistent with Schŕoder et al, 2000.

9. We have added the total amount of fuel consumption and sulfate emissions in 2050 under BL in the manuscript.

10. The base year for comparison in the manuscript is 2006. We have used inventories for 2006 as the base, and 2016, 2026, 2036, 2050 for future projections.

Regards,

Chih-Chieh Chen and Andrew Gettelman

---

## Author Comment (AC3) · 12 May 2016

Dear Reviewer,

We appreciate the suggestions and comments you made on our manuscript. We have modified our manuscript accordingly. Here are the response to your comments.

Major comments:

1. We have added comments in the summary section to address the uncertainties resulting from our treatment of contrails as part of the background clouds. We note that this issue has been discussed in the development of this modeling framework in earlier work (Chen et al., 2012, Chen and Gettelman, 2013 and Gettelman and Chen, 2013.) These are referenced in the text. The major advantage is a self-consistent treatment of contrail water, number concentration and

aerosols in the model hydrologic cycle.

2. We have revised the manuscript to emphasize that the assumption made on the role aviation BC plays can significantly change the results to remind readers that the uncertainties in this regard are high and it is still an active area of research.

3. We have added comments on the sensitivies of the assumed particle size of aviation sulfate aerosols, especially with the higher emissions in 2050.

Minor comments:

1. We have eliminated the original Fig. 6. We have also modified the new Figs.6, 8, 9, 10 to only include perturbation of statistically significance.

2. We have added comments to explain why we ran CAM with specified dynamics instead of the free running mode.

3. We have modified the manuscript accordingly.

4. We have cited the work by Barrett et al. 2010.

5. We have added in the conclusion section on the forcing of aviation $CO_2$ by extrapolating from the 2005 level.

6. We have cited work on the the future aviation emission scenarios in the manuscript (IPCC, 1999, Gierens et al., 1999, Marquart et al., 2003.)

7. We have revised the manuscript as "Sulfate aerosols are emitted to Aitken mode, and is aged into the accumulation mode through coagulation and condensation. Within each mode aerosols are internally mixed and the optical properties reflect this."

8. We have added explanation on why RCP8.5 and 4.5 were selected for the study.

9. It is stated in the manuscript "Future emissions from non-aviation sources and greenhouse gas concentration are based on RCPs for the respective year of the aviation emissions for CAM5-SD simulations." So yes: the different meteorology is taken from simulations with different emissions scenarios.

10. We have added comments to address what is causing the increase of IWP. Indeed, it is mostly due to aviation BC since SC2 shows very similar increase in IWP as in SC1 in which sulfate emission is eliminated.

11. We have acknowleged this and revised the manuscript accordingly.

Regards,

Chih-Chieh Chen and Andrew Gettelman
* * *

---

## Author Response (AR1)

Dear Editor,

We have revised our manuscript according to the comments of three reviewers. Some major changes include: 1) a change to the title of the manuscript to better reflect the scope of the work, 2) added comments in the abstract and the conclusions to address the uncertainties for aviation aerosols, 3) revised figures showing only perturbations of statistical significance (Figs. 6, 8, 9, 10).

The detailed response to the three reviewers' comments is attached in our submission. We would like to highlight that the considerable criticism of the model from Dr. Schumann is mostly related to issues addressed in previous work. Some of the comments show a lack of understanding of how the model works. We have addressed most of these criticisms and description in previously published papers and reference them here. But we do not feel it is appropriate to re-evaluate the model in this manuscript, as it has not changed from previously published work.

However, the concern about uncertainty in the model is valid. We have added significantly to the discussion of uncertainty. This is an improvement.

To address some of the issues raised from previous work, we have added a paragraph to the model description that clearly articulates the philosophy behind the model assumptions are references the model description.

We think this strikes a balance between not repeating earlier work we are not changing, and providing the reader with enough information to highlight uncertainties. We also note that we received similar comments on uncertainty from some of the other reviewers, which we have addressed.

Thank you for your consideration of this reply.

Regards,

Chih-Chieh Chen and Andrew Gettelman

[revised manuscript text omitted]

---

## Author Response (AR2)

Dear Dr. Christopher Hoyle,

We have revised our manuscript according to Dr. Schumann's comments. Specifically, we have modified the abstract to precisely reflect that the 1 cooling effect is due to contrails and aviation aerosols only. We have also followed Dr. Schumann's recommendation to obtain an estimate of the 2050 $CO_2$ radiative forcing based on Sausen and Schumann (2000). Finally, we have also done further analysis on the factor 7 increase for the global contrail radiative forcing in 2050 from the 2006 level and can confirm that it is not an error. Based on our contrail parameterization, there are two sources for contrail ice mass: 1) aviation water vapor emission, and 2) ambient water vapor uptake. The second part can be easily overlooked and thus we have added a sentence in the manuscript to remind readers its importance in extrapolating future contrail radiative forcing. Dr. Schumann also raised a valid point with regard to the factor 6.1 increase in contrail radiative in East Asia, well below the global increase. This is mainly due to how averaging is done. In our definition of East Asia, the blue box in Fig. 6c, it includes area where very light air traffic takes place. If we only include grid cells with heavy air traffic in this region for the regional average calculation, the projected increase in contrail radiative forcing will be significantly higher. A more detailed explanation regarding this concern is included in the response to Dr. Schumann.

Thank you for your acceptance of our manuscript for publication.

Regards,

Chih-Chieh Chen and Andrew Gettelman

Dear Dr. Schumann,

We appreciate your careful review on our manuscript. We have revised our manuscript according to your suggestion. We believe that this has greatly enhanced the clarity of the manuscript. Here are our response to the points you raised:

1. We have revised the abstract according to your recommendation and added the request caveats.

2. We have revised the text according to your suggestion and cite Gierens et. al (1999) and Marquart et al. (2003) in the section on future aviation emission scenarios.

3. The uncertainty estimates have been added to the text.

4. We have checked our calculations and the factor 7 increase of contrail radiative forcing in 2050 is not an error. Based on our contrail parameterization, there are two sources for contrail ice mass upon the formation of contrails: 1) aviation water vapor emission, and 2) ambient water vapor above ice supersaturation within the initial volume of contrails (which is proportional to flight distance). As stated in lines 340–353 of the manuscript, there is minimal spread in contrail radiative forcing between BL and SC1 even though the fuel consumption is reduced by nearly 50% in SC1. This implies that the uptake of background water vapor is an important portion of the contrail ice mass. Therefore, it is essential to take into account the effects of the increase in fuel consumption and flight distance when attempting to estimate the increase of contrail radiative forcing.

   As stated in lines 332–339, the most pronounced increase in contrail radiative forcing in 2050 is in East Asia which is consistent with the projected fuel consumption and flight distance increases. Under the BL scenario in 2050, the fuel consumption is projected to increase by a factor of 7.5 and the flight distance is projected to increase by a factor of 6.2 in East Asia. The contrail radiative forcing in 2050 under the BL scenario within this region, defined in Fig. 6c, is estimated to increase by a factor of 6.1, below the factor 7 for the global increase. Note that the formation of contrails in this region in 2050 under RCP8.5 is substantially suppressed due to the reduction in the frequency of persistent contrail (Fig. 2f). However, within the blue box in Fig. 6c as our definition for East Asia, it includes areas where very light air traffic takes place. If we only include grid cells with heavy air in this region, the projected increase in contrail radiative forcing could be significantly higher than 6.1.

   To illustrate this, we include a figure which illustrates the percentage increase in contrail radiative forcing in 2050 from the 2006 level (panel (a)), and the percentage contribution to the 2050 contrail radiative forcing (panel (b)) for each grid cell, under the BL scenario and RCP8.5. From panel (b), it is clear that the contrail contrail radiative forcing in Central Europe, E. US, the flight corridor over N. Atlantic, Japan, and Indonesia makes the biggest contribution to the global average. It is also seen, in panel (a), that the increase in contrail radiative forcing in 2050 from the 2006 level can exceed a factor of 10 over the flight corridor in N. Atlantic, Japan, and Indonesia. Thus, the radiative forcing in these regions are responsible for the factor 7 increase of the global contrail radiative forcing in 2050.

5. We have followed the methodology of Sausen and Schumann (2000) to obtain an estimate of 2050 $CO_2$ radiative forcing. The manuscript has been revised accordingly.

[Figure]

**(a)** Ratio of contrail radiative forcing: 2050 BL RCP8.5 vs 2006    **(b)** Percentage contribution to the 2050 BL RCP8.5 contrail radiative forcing

Regards,

Chih-Chieh Chen and Andrew Gettelman